# Vascular network-inspired fluidic system (VasFluidics) with spatially functionalizable membranous walls

Yafeng Yu [1], Yi Pan[1,3], Yanting Shen[1], Jingxuan Tian[1,2], Ruotong Zhang [1], Wei Guo [1,2], Chang Li[1] & Ho Cheung Shum [1,2] ✉

In vascular networks, the transport across different vessel walls regulates chemical compositions in blood over space and time. Replicating such trans-wall transport with spatial heterogeneity can empower synthetic fluidic systems to program fluid compositions spatiotemporally. However, it remains challenging as existing synthetic channel walls are typically impermeable or composed of homogeneous materials without functional heterogeneity. This work presents a vascular network-inspired fluidic system (VasFluidics), which is functionalizable for spatially different trans-wall transport. Facilitated by embedded three-dimensional (3D) printing, elastic, ultrathin, and semipermeable walls self-assemble electrostatically. Physicochemical reactions between fluids and walls are localized to vary the trans-wall molecules among separate regions, for instance, by confining solutions or locally immobilizing enzymes on the outside of channels. Therefore, fluid compositions can be regulated spatiotemporally, for example, to mimic blood changes during glucose absorption and metabolism. Our VasFluidics expands opportunities to replicate biofluid processing in nature, providing an alternative to traditional fluidics.

As a biological fluidic system, the vascular network has evolved to self-modulate chemical compositions of blood efficiently through transport across microvessel walls. For instance, capillaries around digestive organs allow glucose to enter to increase blood glucose concentration, while alveolar microvessels enable the gas exchange to decrease carbon dioxide concentration in blood[1,2]. The variable trans-wall molecules among different vessels change fluid compositions from region to region; the high efficiency of trans-wall transport is ensured by the ultrathin ($\approx$1 µm) walls[1,3]. Fabricating biomimetic systems capable of spatially heterogeneous trans-wall transport would promise the spatiotemporal programming of fluid components for biomedical applications[4–7], such as nutrient delivery in artificial tissue constructs and disease modeling in organ-on-chips[8,9].

To date, orchestrating such different trans-wall transports in synthetic channels remains underexplored and challenging. First, channel walls should be ultrathin and semipermeable to allow efficient and selective transport of molecules; to self-adapt to pressure fluctuations during the liquid introduction or extraction, soft walls are preferred over solid walls. Channels with such walls are complicated to build by conventional manufacturing, which typically involves membrane assembly, mold fabrication, substrate casting, and membrane connection to substrates with channel features[9,10]. Encouragingly, the complex process can be simplified into one step by using the emerging embedded 3D printing technique. This technique applies two liquids as printing ink and matrix, extruding the ink within the matrix through a print nozzle. During and after the ink deposition, polymers or nanoparticles pre-dispersed in the ink and matrix will interact on the

[1]Department of Mechanical Engineering, The University of Hong Kong, Pokfulam Road, Hong Kong (SAR), China. [2]Advanced Biomedical Instrumentation Centre, Hong Kong Science Park, Shatin, New Territories, Hong Kong (SAR), China. [3]Present address: Institute of Biomedical Engineering, College of Medicine, Southwest Jiaotong University, Chengdu 610031, China. ✉e-mail: ashum@hku.hk

ink-matrix interface, generating channel-like chambers with liquid cores and self-assembled walls[11–30]. The walls can be soft, semipermeable, and ultrathin (less than 1 μm)[22] to mimic functions of biological soft tissue[16,18,19,29,30]. Second, to control trans-wall transport over different regions, channel walls should be spatially functionalized. Such functionalization can be accessed by removing the matrix, after which self-assembled walls are exposed to the air, allowing further treatment within designated regions. Nevertheless, when isolated from the matrix, the self-assembled channels are too soft to retain 3D-printed architectures at the designated location[30]. Hence, reported channels are typically embedded within the matrix[16–28], only allowing overall homogeneous molecular exchange between the matrix and liquids inside[16–20,25]. In these channels, liquid compositions are simply varied across the whole channel, rather than being adjusted by region-specific reactions to change with a spatiotemporal order.

In this work, we introduce an approach to synthesizing semipermeable soft channels with heterogeneous functions. More specifically, a vascular network-inspired fluidic system (VasFluidics) is introduced, which is functionalizable to adjust fluid components spatiotemporally through trans-wall transport. Facilitated by embedded 3D printing, flexible, thin (1–2 μm), and semipermeable walls are immobilized to a solid substrate, similar to soft tissues supported by bone tissues, to avoid damaging or translocating the printed configurations upon matrix removal. The matrix removal provides access to localize the adjustment of fluids inside, for instance, by confining solutions outside a specific channel for local reagent delivery, or by immobilizing a local channel wall with enzymes to alter flow-through molecules. With these approaches, the trans-wall molecules vary among different regions, facilitating the VasFluidics-based simulation of glucose absorption and metabolism in vascular networks. The fluidic system is potentially exploited to replicate biomolecule synthesis and biofluid dynamics in vivo, serving as a platform for drug discovery and tissue engineering.

## Results

### Fabrication of channels with vascular tissue-like soft walls

Embedded 3D printing is utilized to build channels with self-assembled soft membranous walls. Aqueous solutions of anionic polyacrylamide (APAM) and chitosan are used as printing matrix and ink, respectively. The customized print nozzle is translated under the APAM matrix to deposit chitosan ink onto the bottom of a Petri dish, similar to writing with a pen on paper (Fig. 1a, Supplementary Fig. 1). Both chitosan ink and APAM matrix are shear-thinning liquids, facilitating the controllable ink extrusion and the structural stabilization of deposited inks[31,32] (Supplementary Note 1 and Supplementary Fig. 2). After the ink deposition, coacervates self-assemble on the interface between ink and matrix. Specifically, APAM contains negatively charged groups of carboxylic acid, and chitosan contains positively charged amine groups (Supplementary Fig. 3). The APAM can attract chitosan under electrostatic forces, forming APAM/chitosan coacervates on the ink-matrix interface (Supplementary Fig. 4). The coacervates aggregate into dense and thin membranes when applying appropriate concentrations of chitosan and APAM (Fig. 1b (ii) and Supplementary Fig. 5). During the self-assembly, some chitosan polymers are electrostatically attracted by the surface of the Petri dish, which carries negative charges with a charge density of $-1.50 \pm 0.36\,\mu C\,cm^{-2}$. As a result, the self-assembled membranes adhere to the surface of Petri dish, serving as walls of channel-like hollow chambers (Fig. 1a, b). The thickness of the hydrated wall is adjustable from 1 μm to around 2 μm by changing the polymer concentrations, the pH, and the membrane assembly time (Supplementary Fig. 6). The thickness is comparable to that of capillary walls in the human body, which is around 1 μm[1,3]. Moreover, the self-assembled membranes are soft and elastic with Young's modulus of 1–9 kPa (Supplementary Fig. 7a), similar to that of some biological soft tissues, such as skin, spleen, and pancreas[33]. Thus,

the channel-like chamber can deform under external force and resume the original shape upon removal of the force (Fig. 1c, Supplementary Movie 1). The membranes can be even softer in saline, acid, and alkaline conditions (Supplementary Fig. 7b). The flexibility of the membranes is presumably attributed to the long polymer chains, which endow the formed membranes with abundant chain entanglements[34,35]. Moreover, the chamber walls can heal themselves. Once the walls are punctured, chitosan ink inside the chamber can react with the APAM matrix outside to generate new chitosan/APAM complexes to "heal" the walls (Supplementary Fig. 8).

The printed chambers can be engineered into various architectures with different sizes by adjusting printing parameters. The size of a single chamber is adjustable by applying different printing speeds and ink flow rates (Supplementary Note 2, Supplementary Figs. 9–12). Various chamber structures are obtained by further programming the printing path, including a spiral, a grid, a tree-like branch, and a vascular-shaped network (Fig. 1d, e). The chambers remain intact with designed architectures when removing the printing matrix and ink (Fig. 1e, Supplementary Fig. 13, and Supplementary Fig. 14). The printed structures can serve as fluidic channels when perfused with liquids inside (Fig. 1e, Supplementary Movie 2). Due to the elasticity of the walls, the fluidic channels can be inflated or collapsed to alter intracavity volume in response to the changes in liquid volume inside, similar to the way in-vivo vessels change sizes under different blood pressure. Specifically, the channel with soft walls collapses when liquids inside flow out; by contrast, the channel inflates as the liquid volume inside increases (Fig. 1f, Supplementary Movie 3). Notably, the walls will detach from the substrate when the flow rate of liquid inside exceeds the allowable flow rate. The range of allowable flow rates in different-sized channels is listed in Supplementary Fig. 15 and analyzed in Supplementary Note 3 and Supplementary Fig. 16. The fluidic channel walls cannot self-heal after the removal of printing ink and matrix (Supplementary Fig. 17).

### Localized trans-wall transport of specific molecules

Similar to blood vessel walls, walls of the printed channels have selective permeability. The selective trans-wall transport is visualized with aqueous solutions of fluorescent dextran molecules (Fig. 2a). Molecules with hydrodynamic radii ($R_h$) less than or equal to 1.9 nm can penetrate the wall, whereas molecules with a large $R_h$ ($\geq 3.0$ nm) cannot. The dimensions of some of the solutes used in this study are listed in Supplementary Table 1. Such size-selectivity indicates that the channel walls can block various possible contaminants from entering channels, such as floating or suspended solids, microorganisms, and even viruses (the minor diameter of viruses is 20–30 nm). The permeability of channel walls varies under different conditions; for example, it increases in saline and acid environments, and decreases in alkaline environments (Supplementary Fig. 18).

By localizing the trans-wall transport of specific molecules, liquid components within the channel can be spatiotemporally regulated (Fig. 2b, c, Supplementary Movie 4). Solution of a few microliters to a few milliliters can be directly deposited on or next to the channel wall after removing the printing matrix and exposing the channel to air (Supplementary Fig. 19). Thus, the solute molecules locally exchange between placed drops and liquids inside, varying compositions of the downstream liquids (Fig. 2b). To provide a proof-of-concept demonstration, a Y-shaped channel is printed, exposed to the air, and infused with aqueous solutions of dyes capable of penetrating the walls (Fig. 2c(i)). When depositing a water drop on the channel, local dyes inside are extracted and diffused into the droplet (Fig. 2c(ii)). Similarly, the introduction of methylene blue (MB, a blue dye) into the channel is localized by attaching an MB drop to the channel (Fig. 2c(iii)). Since droplets can be deposited at random positions and times, the localized molecule introduction and extraction indicate the real-time regulation and monitoring of liquid components. For instance, at any time point,

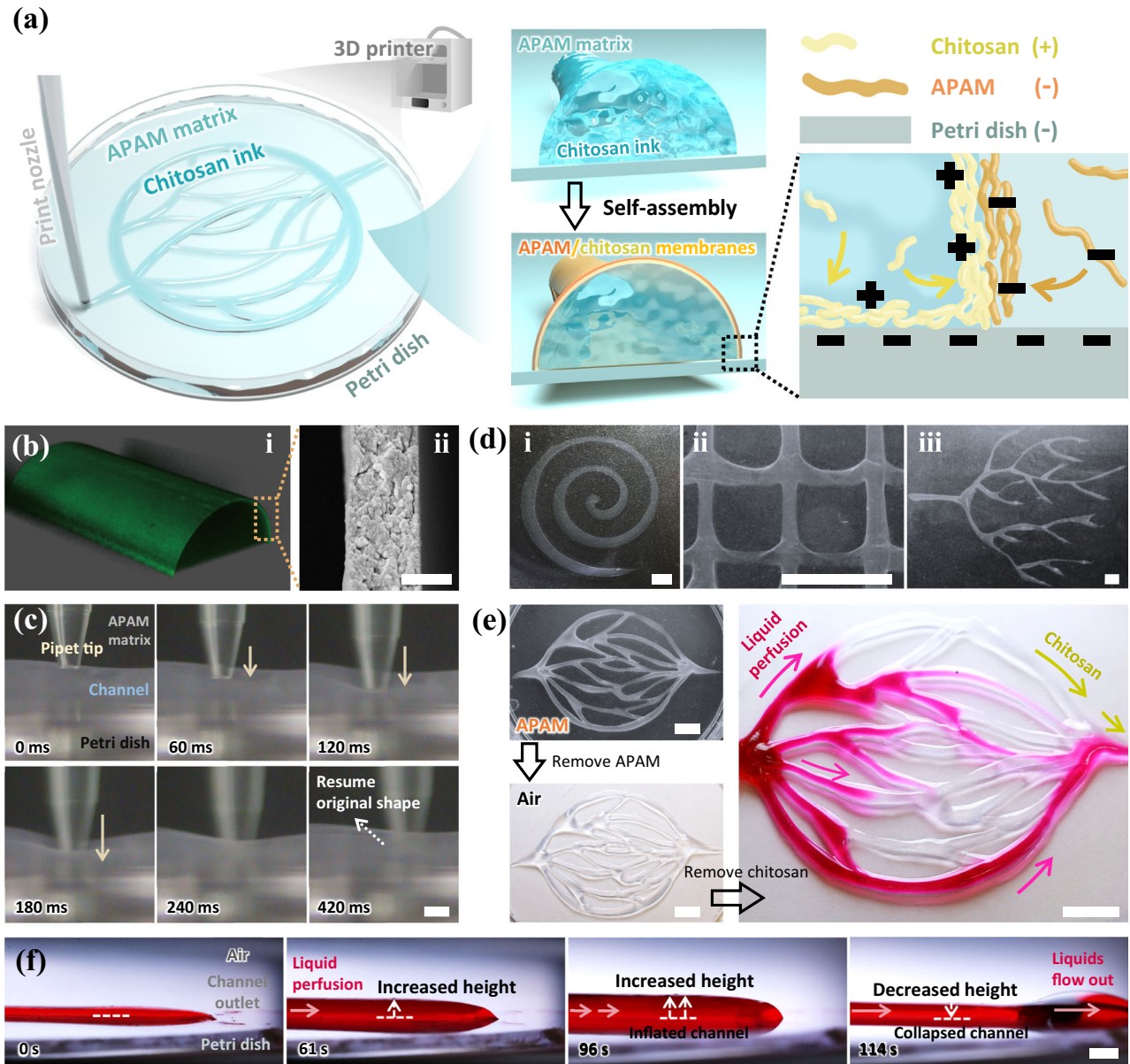

**Fig. 1 | Embedded 3D printing of channels with self-assembled soft membranous walls. a** Schematic showing the printing and the self-assembly process of channels. The print nozzle is translated within anionic polyacrylamide (APAM) matrix to deposit chitosan ink against the bottom surface of a Petri dish. Driving by the electrostatic forces between chitosan, APAM, and Petri dish carrying charged groups, channel-like chambers generate with self-assembled chitosan/APAM membrane walls, which are attached to the dish surface. **b** (i) Confocal laser scanning microscope image showing self-assembled chambers with hollow liquid cores and thin walls. 0.02 wt% fluorescein isothiocyanate-labeled chitosan (FITC-chitosan) is mixed in the chitosan ink for visualization. (ii) Scanning electron microscope image showing the ultrathin and dense self-assembled wall. The membranous wall is freeze-dried before observation. The scale bar is 500 nm. **c** Soft and elastic channel walls deform under external force and resume the original shape upon removal of the force. The external force is applied by using a pipette tip. The scale bar is 1 mm. **d** Photographs of 3D-printed channels with arbitrary architectures, including (i) a spiral-shaped chamber, (ii) a grid-shaped chamber, and (iii) a branched network. Scale bars here are 5 mm. **e** Removal of matrix and infusion of liquids into a vascular network-shaped channel. The channel remains intact during the removal of matrix and ink. The channel is exposed to the air when infused with rhodamine 6 G (R6G, a red dye) aqueous solution. The scale bar is 1 cm. **f** Time-series optical images showing intracavity volume of the channel can alter in response to the changed liquid volume inside. The channel with soft walls collapses as liquids inside have been partially removed via the channel outlet. The height of the channel increases from 780 to 1560 μm when the liquid perfusion rate increases from 0 to 3 mL h$^{-1}$. The height decreases to 970 μm when liquids inside partially flow out of the channel. The liquid for perfusion is an aqueous RhB solution. The scale bar is 2 mm.

the concentration of downstream small molecules ($R_h \leq 1.9$ nm) in a specific branch can be increased or decreased by attaching water or solution droplets on channels; small molecules ($R_h \leq 1.9$ nm) at specific regions can be extracted to monitor whether intermediate products are generated as desired.

Moreover, as the trans-wall molecules penetrate gently, the alteration of downstream components through localized trans-wall

transport can persist over a programmable period. For validation, we track the MB concentration of the internal flowing liquids, which varies continuously over 20 min after we simply deposit an MB droplet beside the upstream channel (Fig. 2d and Supplementary Fig. 20). The availability of the MB can be programmed by varying the MB concentration of the droplet. For instance, after introducing a 0.1 wt% MB droplet, the downstream MB becomes undetectable after 30 min; in

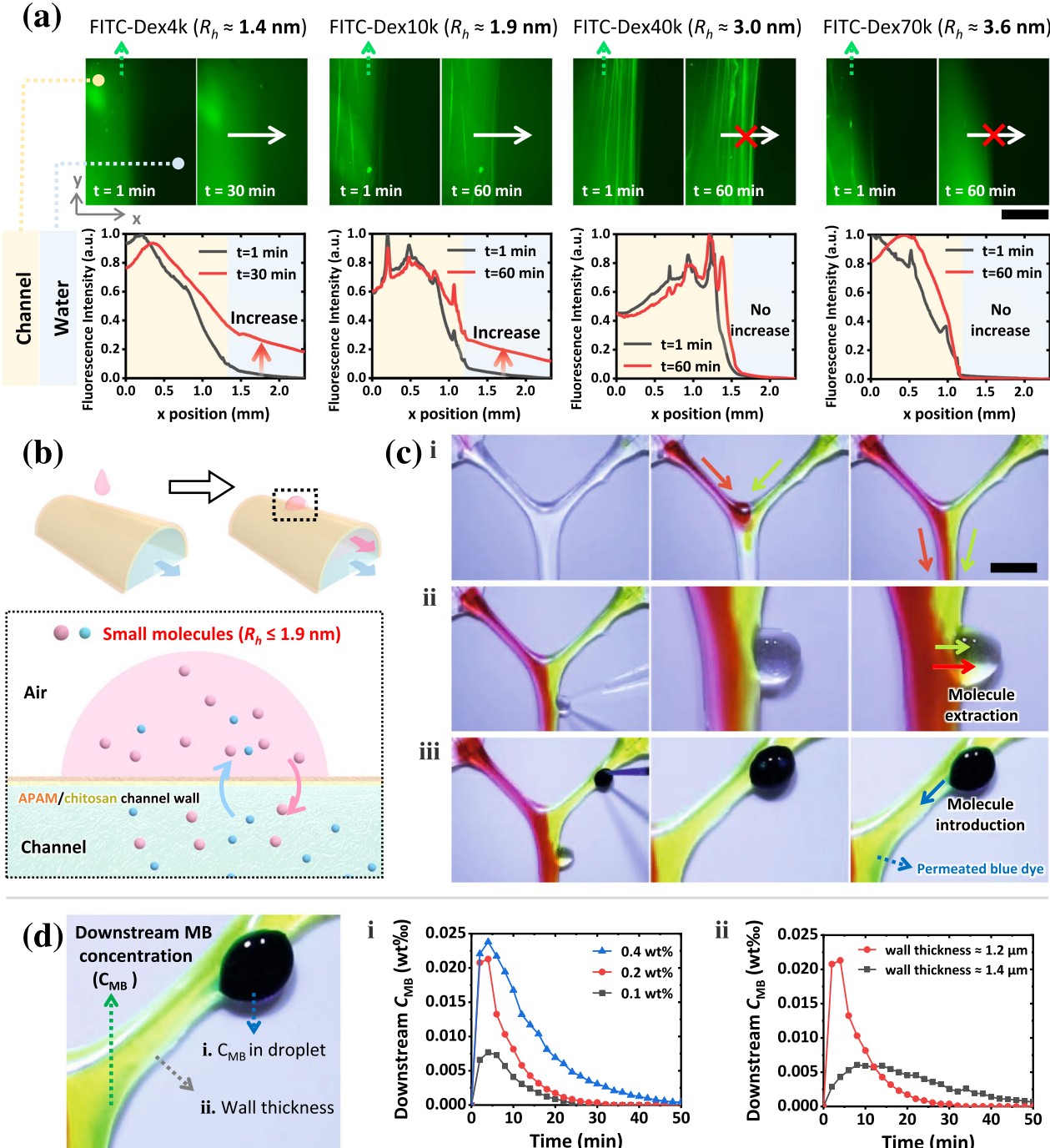

**Fig. 2 | Spatiotemporal regulation on fluid compositions by localizing trans-wall transport of specific molecules. a** Fluorescence microscope images showing the membrane selectivity for different-sized molecules. Aqueous solutions of fluorescent molecules with different hydrodynamic radii ($R_h$) are infused into channels at a flow rate of 0.5 mL h⁻¹. Channel walls under view are immersed under 200 μL deionized water. Fluorescein isothiocyanate-labeled dextran with a molecular weight of 4000 g mol⁻¹ (FITC-Dex4k, $R_h \approx 1.4$ nm)[41] and 10,000 g mol⁻¹ (FITC-Dex10k, $R_h \approx 1.9$ nm)[41] are observed passing through the channel wall within 30 min and 60 min. Trans-wall transport of fluorescein isothiocyanate-labeled dextran with a molecular weight of 40,000 g mol⁻¹ (FITC-Dex40k, $R_h \approx 3.0$ nm)[41] or 70,000 g mol⁻¹ (FITC-Dex70k, $R_h \approx 3.6$ nm)[41] is not observed within 60 min. Relative fluorescence intensities in images in arbitrary units (a.u.) are analyzed with MATLAB. The scale bar is 1 mm. **b** Localized trans-wall exchange between liquids inside and the solution deposited outside. **c** Spatiotemporal regulation of fluid components via the localized trans-wall introduction and extraction. (i) A Y-shaped channel is perfused with rhodamine 6 G (red dye) and T/Thioflavin T (yellow dye) aqueous solutions. (ii) Dyes at random positions and times can be extracted by depositing one deionized water droplet next to the channel. (iii) Methylene blue (MB, a blue dye) can be introduced into the channel at random positions and times by placing an MB droplet next to the channel. The scale bar is 1 cm. **d** Temporal regulation of MB concentration inside the channel via localized trans-wall MB transport. The channel is perfused with deionized water at a flow rate of 3 mL h⁻¹. After placing a 5 μL MB droplet beside the channel, liquids inside are collected 1 cm downstream from the droplet position every 2 min to analyze the MB concentration inside channel. MB concentrations with different dynamics are presented when (i) different concentrated MB droplets are placed beside the channel, or (ii) channels with different wall thicknesses are applied. Wall thickness here refers to the thickness of hydrated membrane walls. Source data are provided as a Source Data file.

contrast, if a 0.4 wt% MB droplet is used instead, it takes 50 min for the MB to be undetectable (Fig. 2d(i)). Apart from the programmability in duration, the time for the peak concentration of MB can also be controlled. For example, the peak time of the downstream MB concentration can be postponed from 4 min to 8 min by increasing channel wall thickness from 1.2 to 1.4 μm (Fig. 2d(ii)). Such temporally programmable introduction of new molecules can be exploited to control drug dynamics in in-vitro models, replicating or exploring human response to new drugs. For instance, by programming drug permeation into channels, the absorption of sustained-release drugs into bloodstream can be simulated. During the simulation, the peak time of drug concentration can be postponed when utilizing channels with thicker walls; the duration of effective drug concentration can be extended by repeatedly introducing the drug inside, which is also demonstrated with MB (Supplementary Fig. 21).

## Local immobilization of enzymes on channel walls

Besides physically allowing specific molecules to pass by, the walls can chemically alter fluid compositions when modified with functional materials. The functional materials should have large sizes (radius or $R_h \geq 3.0$ nm) and carry electrostatic charges, such as some polymers, particles, and proteins. When placing corresponding solutions outside channels, these materials can be electrostatically attracted to the walls without penetration. As a result, the channel walls are modified with an extra layer of functional materials (Fig. 3a). Such modification on walls is demonstrated with glucose oxidase (GOx) enzyme and horseradish peroxidase (HRP) enzyme, both of which have $R_h$ larger than 3.0 nm[36,37] and cannot penetrate channel walls (Supplementary Table 1). At neutral pH (pH = 6–8), GOx is negatively charged, and HRP is positively charged. Therefore, positively charged FITC-labeled HRP (FITC-HRP, green) can be attracted by the negatively charged APAM and immobilized on the channel wall (Fig. 3a (iii)); negatively charged and RhB-labeled GOx (RhB-GOx, red) can also be adsorbed onto the wall via layer-by-layer assembly[21,38] (Fig. 3a (iv)).

The extra-coating can spatially distribute on a single channel with variable coating amounts, which is demonstrated with enzymes as an example. Different enzymes can be co-immobilized on the channel walls, separately or overlappingly (Fig. 3b and Supplementary Fig. 22). A channel is modified with two RhB-GOx-coated regions, one HRP-coated region, and an RhB-GOx and FITC-HRP co-coated region, as illustrated in Fig. 3b(i), (ii). Among these regions, the immobilized enzyme amount can be varied, for instance, by treating local walls with different concentrations of enzyme solutions. Different fluorescence intensities are exhibited in regions treated with different concentrated RhB-GOx solutions, showing that the immobilized enzyme amount is tunable (Fig. 3b(iii) and Supplementary Fig. 22c). These immobilizations with different spatial distribution and enzyme loading amount result in the functional heterogeneity of channel walls. For instance, regions thickened with multiple coacervate layers can transport molecules across the wall at a slower rate; regions coated with functional materials, such as enzymes, can speed up chemical reactions inside channels.

## Spatially arranged physiochemical reactions between fluids and VasFluidic device

In the same VasFluidic device, different regions can be assigned simultaneously for both trans-wall transport and enzyme immobilization. As a demonstration, we present a VasFluidic channel in which 4 downstream regions are immobilized with enzymes, and 4 upstream regions are designated for dye introduction (Fig. 4 and Supplementary Movie 5). The 4 downstream regions are treated with a mixed solution of FITC-HRP and negatively charged green fluorescent microparticles for enhanced fluorescence (Fig. 4b(i)). The predetermined 4 upstream locations are attached with R6G (red) and fluorescein sodium (green) solutions, penetrating the walls and modulating the fluid compositions

in corresponding downstream channel branches. (Fig. 4b(ii-iv)). The location and number of regions for dye introduction (or enzyme immobilization) can be varied on demand, as shown in Supplementary Fig. 23.

When functionalizing VasFluidic channels with these compartmentalized domains, various region-specific reactions with fluids can proceed under spatiotemporal control. As a proof of concept, a VasFluidic channel is engineered for a multistep cascade reaction to simulate the control over glucose in the vascular network (Fig. 5a): In the vascular network, vessels in digestion organs do not directly take in starch before it is degraded into glucose. Absorbed glucose will be partially degraded by endothelial cells on vessel walls, during which carbon dioxide ($CO_2$) is produced. The cumulated $CO_2$ in blood will finally escape through vessel walls, partially exhaled by the lungs. A similar process is executed in a VasFluidic device (Fig. 5b): The upstream channel selectively takes in glucose from the attached droplets; the glucose is degraded into smaller molecules in the midstream channel by the action of pre-immobilized enzymes. $CO_2$ is further generated and exhaled from the permeable walls in the downstream channel.

Specifically, the upstream channel selectively intakes glucose (Fig. 5c). When a solution mixture of starch and glucose is deposited above the channel, the channel block starch (granules radius ≈ 2.5–10 μm)[39] from the outside, as shown in Fig. 5c(i). In contrast, around 94% of glucose ($R_h \approx 0.4$ nm)[40] in the starch-glucose solution permeates into the channel within 30 min (Fig. 5c(ii)). During the glucose permeation, the downstream glucose concentration presents temporal fluctuation (Fig. 5c(iii)). The glucose variation is similar to that in the vascular network during food digestion, where blood glucose concentration increases first and then gradually decreases. Here, the glucose introduction is limited to around 1 hour to restrict the duration of the following glucose degradation (Fig. 5c(iii)).

The absorbed glucose is partially degraded while flowing through the midstream channel, in which two separated regions are immobilized with GOx and HRP, respectively (Fig. 5d). Since glucose can pass across membrane walls to meet enzymes on the external wall, the GOx-coated region oxidizes glucose into D-gluconolactone and hydrogen peroxide ($H_2O_2$), and the produced $H_2O_2$ is further converted into $H_2O$ in the following HRP-coated region (Fig. 5d(i)). This bi-enzymatic cascade reaction is indicated by Amplex Red, which is converted into resorufin to emit red fluorescence during the HRP-mediated $H_2O_2$ catalysis (Fig. 5d(ii)). Notably, the red fluorescence is present only when fluids flow through the HRP-coated region, indicating region-specific HRP-mediated catalysis. As for channels immobilized with GOx or HRP only, the bi-enzymatic cascade reaction does not occur (Supplementary Fig. 24).

The glucose degradation is followed by downstream trans-wall emission of $CO_2$, similar to the $CO_2$ exhalation by lungs after nutrient absorption (Fig. 5e). Since $CO_2$ exhaled by lungs is generated from accumulated bicarbonate ($HCO_3^-$) in blood, we produce the downstream $CO_2$ with sodium bicarbonate ($NaHCO_3$) and hydrochloric acid (HCl). $NaHCO_3$ and HCl are perfused and designed to meet in the downstream junction, as shown in Fig. 5b and Fig. 5e(i). Setups to detect the trans-wall $CO_2$ are presented in Supplementary Fig. 25. The $CO_2$ concentration surrounding the channel remains unchanged when infusing the channel with HCl only. However, the $CO_2$ concentration increases when perfusing the channel with $NaHCO_3$ and HCl simultaneously; this is attributed to the trans-wall $CO_2$ escaping from the channel inside (Fig. 5e(ii)). By expelling $CO_2$ via trans-wall transport, the pressure, pH, and gas contents of the flowing fluids inside are regulated.

In the biomimetic VasFluidic device, fluid compositions are programmably regulated over space and time, via the various region-specific reactions (Fig. 5b–e). The dynamics of fluid compositions in VasFluidic channels can be even more complex, for instance, by

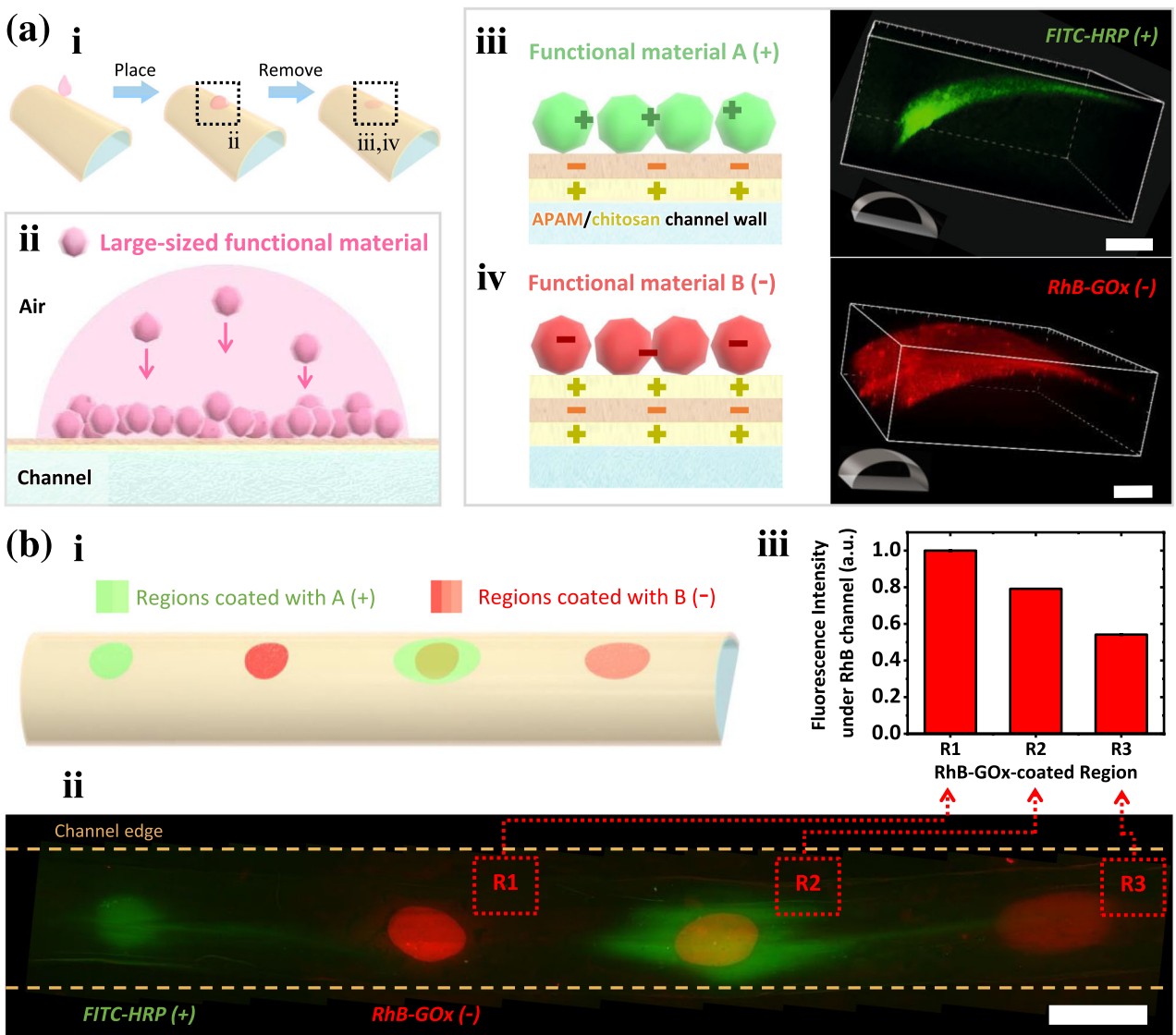

**Fig. 3 | Spatial immobilization of functional materials on channel walls. a** (i, ii) Schematic showing the localized immobilization of functional materials on channel walls by attaching solutions to the channel. The large-sized solutes (radius or $R_h \geq 3.0$ nm) carrying electrostatic charges cannot penetrate membrane walls but can be electrostatically immobilized on the membranes. (iii, iv) Schematics showing the immobilization of negatively or positively charged solutes on channel walls; Confocal laser scanning microscope images showing channel walls immobilized with the positively charged FITC-labeled horseradish peroxidase (FITC-HRP, green) or negatively charged RhB-labeled glucose oxidase (RhB-GOx, red). Scale bars are 200 μm. **b** (i) Schematic and (ii) fluorescence microscope images showing the spatially arranged immobilization of the two enzymes on the channel walls. RhB-GOx-coated regions are visualized under RhB channel of the microscope, and FITC-HRP-coated regions are visualized under FITC channel of the microscope. Fluorescence images under RhB channel and FITC channel are collected, respectively, and merged into one. To obtain the three RhB-GOx-coated regions, region R1 is treated with 0.5 μL, 10 mg mL$^{-1}$ RhB-GOx solution; region R2 is treated with 0.5 μL, 4 mg mL$^{-1}$ RhB-GOx solution; region R3 is coated with 1 μL, 1 mg mL$^{-1}$ RhB-GOx solution. The left HRP-coated region is treated with 0.5 μL, 1 mg mL$^{-1}$ FITC-HRP aqueous solution, and the right region is coated with 1 μL, 4 mg mL$^{-1}$ FITC-HRP aqueous solution. The scale bar is 2 mm. (iii) RhB-GOx-coated regions treated with different concentrations of RhB-GOx solutions have different fluorescence intensities. The relative fluorescence intensities in arbitrary units (a.u.) are obtained by analyzing fluorescence microscope images under RhB channel. Source data are provided as a Source Data file.

designing channels with complicated structures to connect several cascade fluid reactions in parallel.

## Discussion

We have introduced a vascular network-inspired fluidic system (VasFluidics) with soft tissue-like membrane walls. The VasFluidic channels are spatially functionalizable to react with fluids at compartmentalized domains, thus capable of regulating fluid compositions in a spatiotemporal manner. Facilitated by embedded 3D printing, polymers are self-assembled to form membranous walls of fluidic channels. The walls are flexible, ultrathin, semipermeable, and immobilized to the Petri dish substrate to remain printed configurations during the removal of printing ink and matrix. By depositing solutions or immobilizing enzymes on separated regions of channel walls, VasFluidic channels are functionalized for different region-specific flow chemistry: Some regions physically allow specific molecules to pass across walls, while some chemically alter fluid compositions. Thus, the transwall molecules vary over different regions, resulting in the spatiotemporal variation of fluid compositions in VasFluidic channels. Such space- and time-varying fluid components are ubiquitous in natural fluidic systems but challenging to implement in existing synthetic fluidics. Hence, we envision our VasFluidics will extend the functions of the existing fluidic devices. Although liquid composition distribution in VasFluidic channels may also be influenced by the non-uniformity of

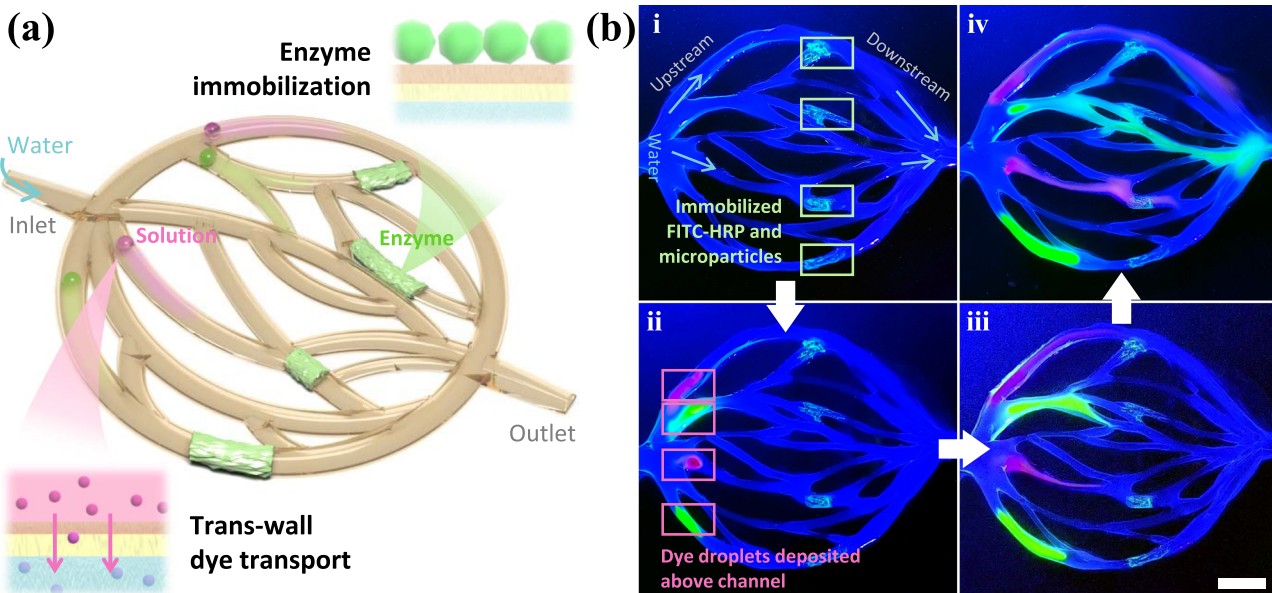

**Fig. 4 | Spatial arrangement of different regions for trans-wall dye transport and enzyme immobilization. a** Schematic of a vascular network-shaped VasFluidic device with 4 downstream regions immobilized with enzymes and 4 upstream regions taking in dyes via trans-wall transfer. Corresponding optical images of the VasFluidic device are also presented: **b** A VasFluidic channel being perfused with water inside and exposed under 405 nm light. (i) 4 regions on the downstream channels coated with FITC-HRP and 1 μm negatively-charged green fluorescent particles. The fluorescent particles are used for enhancing the fluorescence in enzyme-immobilized regions. (ii-iv) 4 upstream regions attached with dye solutions, where dye molecules are introduced into the channel to change fluid compositions in different channel branches. The dyes are R6G (red) and fluorescein sodium (green). The scale bar is 1 cm.

the channel walls, which are partially attached to the substrate, such influence could be reduced by the extra modification on the inner surface of walls, for instance, via layer-by-layer assembly[21,38]. Besides, the fast-evolving 3D printing may further push the envelope in the geometry complexity of VasFluidic devices, such as building ultra-fine channels with intricate organization, for replicating the flow in blood capillaries with a diameter of 7–9 μm. Overall, we believe our VasFluidics can pioneer complex fluid manipulation or even revolutionize the way in fluid processing, applicable to areas including but not limited to biological fluid mechanics, biomolecule synthesis, and drug screening.

## Methods

### Printing ink and printing matrix preparation

Purified deionized water (Direct-Q 5UV-R, Merck), anionic polyacrylamide (APAM, average molecular weight (Mw) ≥3,000,000 g mol⁻¹, ≥85.0%, purchased from Hushi, China), chitosan (>400 mPa·s, purchased from Aladdin, China), acetic acid (Molecular weight = 60.05, ≥99.07%, purchased from Acros Organics, Belgium), sodium hydroxide (NaOH, Mw = 40.00, 97%, purchased from Aladdin, China) were used for solutions preparation. Chitosans with shorter chains (<200 mPa·s, purchased from Aladdin, China) were not recommended for ink preparation, since we experimentally found that the interfacial chitosan/APAM assemblies would not be robust enough to lock the short-chained chitosans inside, forming chambers with formless walls. To prepare the printing ink, chitosan was dissolved in water-diluted acetic acid, and the pH was adjusted with an aqueous NaOH solution. A pH meter (PH550 Benchtop pH Meter, Oakton) was used to prepare the chitosan solution with a given pH value. APAM was dissolved in deionized water with gentle shaking overnight for complete dissolution. Chitosan solutions and APAM solutions were utilized within 3 days after preparation. Unless otherwise specified, the chitosan printing ink was 1 wt% chitosan solution (pH ≈ 6.0–6.5), and the APAM printing matrix was 1 wt% APAM solution. 2 wt% APAM matrix was used when printing the vascular network-shaped channel in Figs. 1e, 4. The rheological properties of the APAM and chitosan solutions were measured by a commercial rheometer (MCR 320, Anton Paar).

### Surface charge density measurement of Petri dishes

The Petri dishes were cut by a clean scissor into small pieces with areas of approximately 10 cm². The exact area was measured after the surface charge density measurement. The small pieces were transferred by a clean tweezer into a Faraday cup. The Faraday cup was connected with a programmable electrometer (6514, Keithley Instruments Model) under the charge measurement model. For measurements of the same petri dish type, pieces were collected from 3 petri dishes, and each piece was measured 3 times. The surface charge densities were calculated by dividing the measured charge value by the measured area of samples.

### Embedded printing of VasFluidic channels

Both tissue culture-treated dishes (100 mm × 20 mm Style or 60 mm × 15 mm Style, Polystyrene, Tissue culture treated, Sterile, Labserv, Thermo Fisher Scientific) or non-treated dishes (101VR20, 90 mm, Polystyrene, Non-treated, Non-sterile, Thermo Fisher Scientific) can be used as substrates for channel printing. Although the surface charge density of tissue culture-treated dishes (−1.81 ± 0.53 μC cm⁻²) is higher than that of untreated dishes (−1.50 ± 0.36 μC cm⁻²), the untreated ones are preferred due to their surface hydrophobicity. When using dishes with hydrophobic surfaces, the liquid matrix would be more easily removed in subsequent operations. The untreated polystyrene dishes were used in all experiments presented in this study. A commercially available 3D printer was used in all printing experiments (Ultimaker 2 +, Netherlands). The customized print nozzle was one commercially available 0.5–10 μL pipette tip (0.5–10 μL Clear Tips, Nonpyrogenic, DNAse/RNAse Free, ExCell Bio) connected to a syringe via PTFE tubing. The syringe was prefilled with chitosan printing ink and mounted on a syringe pump (LSP02-2B, LongerPump). An illustration of the setups is presented in Supplementary Fig. 1. The build plate of the 3D printer was leveled using miniature bull's-eye spirit levels, and the dish was horizontally

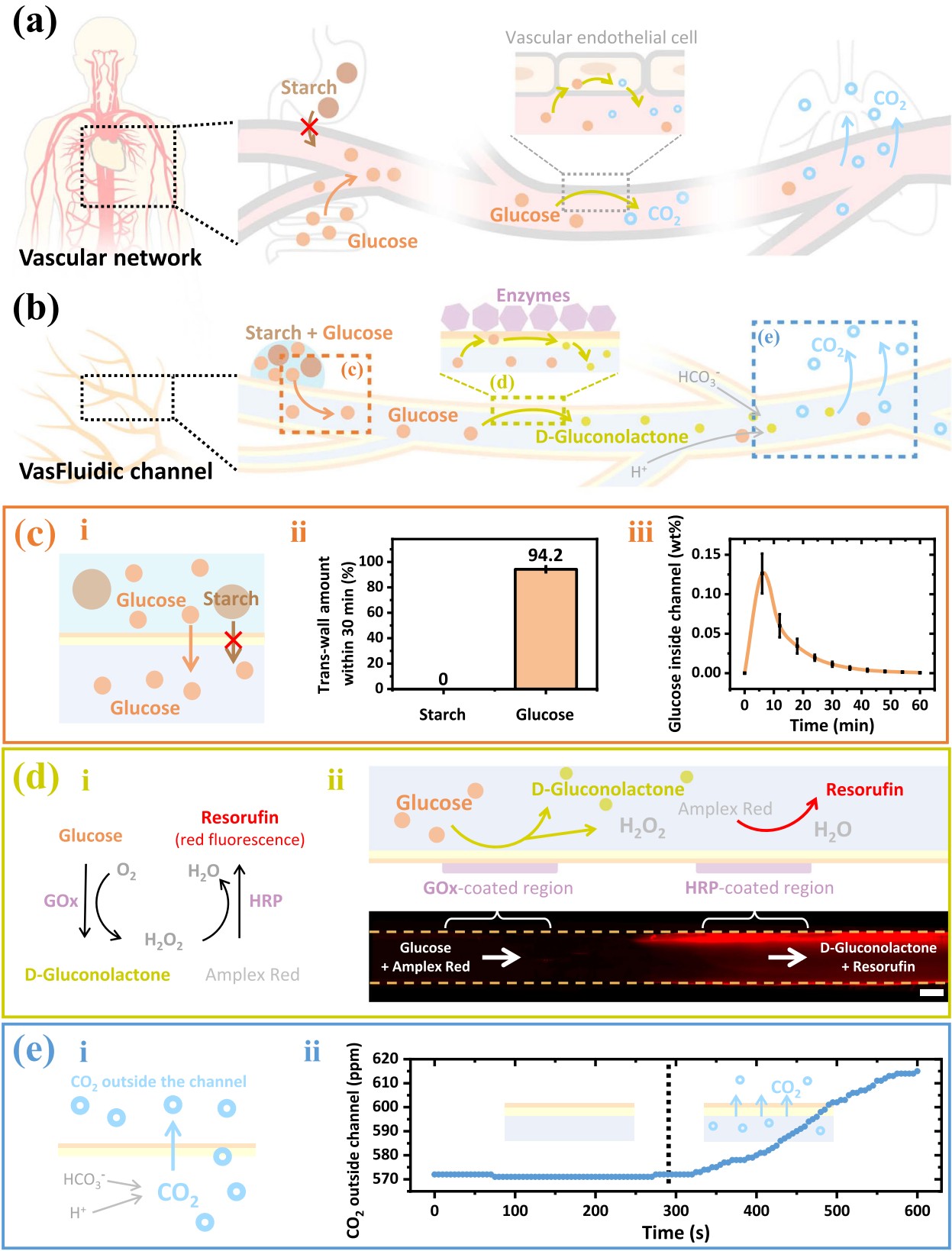

placed on the build plate. We set the zero position by lowering the print nozzle until it reached the bottom surface of the Petri dish. The customized print nozzle could experience resistance and slightly bend when sliding the Petri dish around. The Petri dish was then poured with APAM printing matrix solution, and the printing matrix solution had a height of more than 1 cm. The Petri dish containing the printing matrix

was then fixed on the build plate with adhesive tape to avoid possible movement during further printing. During the ink deposition, the print nozzle should be immersed under the printing matrix, keeping touching the bottom surface of the Petri dish to deposit printing ink. If interconnected channels were required, vertically puncture the channel printed within 5 min and then print the branch channel. APAM and

**Fig. 5 | Spatially arranged physicochemical reactions between fluids and VasFluidic channel walls. a** Schematic showing a cascade process of glucose absorption, glucose degradation, and carbon dioxide ($CO_2$) exhalation in in-vivo vascular networks. **b** Schematic showing a VasFluidic channel with compartmentalized functional domains, which enable glucose absorption, glucose degradation, and $CO_2$ exhalation, respectively. **c** (i) Schematic showing the upstream channel takes in glucose selectively from a solution mixture of starch and glucose. The branch channel is perfused with deionized water at a 1.5 mL h⁻¹ flow rate. 2 μL starch-glucose aqueous solution containing 20 wt% starch and 20 wt% glucose is attached to the upstream channel. Liquid samples are then collected in the 1 cm downstream channel every 6 min. By analyzing glucose and starch concentrations in collected samples with corresponding assay kits, respectively, (ii) the trans-wall amount of glucose and starch within 30 min and (iii) the real-time glucose concentration of liquids inside the channel can be derived. **d** (i) Schematic showing an enzyme-mediated cascade reaction to decompose glucose, which can be visualized by a red fluorescence probe. (ii) Schematic and (iii) fluorescence microscope images showing the midstream decomposition of glucose into D-gluconolactone. The midstream channel with a GOx-coated region and an HRP-coated region converts glucose and Amplex Red into D-gluconolactone and resorufin, respectively. The scale bar is 1 mm. **e** (i) Schematic showing the downstream exhalation of $CO_2$ via trans-wall transport. The $CO_2$ is produced with sodium bicarbonate ($NaHCO_3$) and hydrochloric acid (HCl). (ii) Real-time $CO_2$ concentration outside the downstream channel. The $CO_2$ concentration remains unchanged when the channel is perfused with HCl aqueous solution only (0–300 s). The $CO_2$ concentration increases as we perfuse the channel with HCl and $NaHCO_3$ aqueous solutions simultaneously (300–600 s). Source data are provided as a Source Data file.

chitosan polymers could self-assemble on the ink-matrix interface around the punctured position. We used the printing speed of 100–800 mm min⁻¹ and the ink flow rate of 3–10 mL h⁻¹ to print channels with different sizes. Unless otherwise specified, we waited 20 min after printing for APAM/chitosan complex assembly, which was ended by removing the APAM matrix. The assembly time of the vascular network-shaped channel in Fig. 1e was more than 1 h. For the purpose of examining the dye concentration variation, the assembly time of the channel in Fig. 2d was shortened to 15 min, and thinner membrane walls were resulted. All printing processes were done at room temperature.

### Liquid perfusion into the self-assembled channel

A commercially available stainless-steel needle with an external diameter of 250 μm or 600 μm was connected to a syringe via PTFE tubing. The syringe was prefilled with deionized water (pH ≈ 7) and mounted on a syringe pump. After checking there were no bubbles in the needle-tubing-syringe connection device, we inserted the needle into the channel inlet. 20 wt% gelatin aqueous solution (37 °C) was placed around the channel inlet for sealing, which could solidify at room temperature to avoid leakage from the channel inlet. The gelatin solution was prepared by dissolving gelatin (Gelatin, CP, purchased from Aladdin, China) in deionized water under 50 °C. To wash out the chitosan ink inside the channel, we perfused the channel with deionized water at a flow rate of 1–3 mL min⁻¹ for more than 10 min. After the removal of chitosan ink, water in the syringe can be replaced for solution perfusion as desired. During or before the liquid infusion into channels, the channel walls should be kept hydrated. Liquids cannot be injected into the dried channels (Supplementary Fig. 26). The channels can be stored for at least 7 days if the channel walls are kept hydrated (Supplementary Fig. 27).

### Optical and fluorescence microscope imaging

In all experiments, the vertical views of printed channels were collected with a digital camera (Canon EOS 70D) or by an inverted fluorescence optical microscope (Leica microscope). The cross-sectional views of channels were collected by using a confocal laser scanning microscope (TCS SP8, Leica). The cross-sectional views of polymer aggregation on the liquid interface were observed with another confocal laser scanning microscope (Eclipse Ti2-E, Nikon). For visualization under the confocal laser scanning microscope, channels were printed by ink pre-mixed with 0.02 wt% fluorescein isothiocyanate-labeled chitosan (FITC-chitosan, customized by Xi'an Ruixi Biological Technology Co. Ltd, China).

### Young's modulus measurement of hydrated membranes with an AFM microscope

0.5 mL chitosan solution was placed on a Petri dish surface. The dish was tilted so that the solution could cover an area over 3 cm². 5 mL APAM solution was poured on top of the chitosan solution and left standing for a given time period for self-assembly. The APAM and chitosan solutions were then refreshed with deionized water at least 10 times. After that, the membranes were immersed under under deionized water, saline solutions, HCl solutions, or NaOH solutions, separately. After letting membranes stand for at least 3 h, the membranes could physically attach to the Petri dish surface. An atomic force microscope (Nano Wizard, JPK Instruments) was used to obtain force-distance curves. The indentation test was conducted by using a scanning probe with a plateau tip (SD-PL2-CONTR-10, Silicon-SPM-Sensor with plateau tip, plateau diameter: 1.8 μm ± 0.5 μm, force constant: 0.02–0.77 N m⁻¹, NANOSENSORS). During the test, both membranes and AFM tip were immersed in water or aqueous solutions. For each sample, more than 8 test points were selected, and each test point corresponds to 3–5 test results. Regions without wrinkles or bubbles under the field of the optical microscope were selected. The resulting force vs. displacement curves were analyzed by JPK Data Processing Software (Version 6.1.159, Bruker) to determine Young's modulus of samples. Hertz/Sneddon Model was exploited as the fitting model, and the unknown Poisson ratio was set to 0.5. The order of magnitudes of obtained results did not vary as the Poisson ratio changed from 0.5 to 0.1.

### Thickness measurement of freeze-dried membranes with a SEM microscope

We referred to the reported research[22] to prepare the self-assembled membranes for thickness measurement. The APAM solution was poured on top of the chitosan solution and waited for a given time for self-assembly. The APAM and chitosan solutions were then refreshed with deionized water at least 10 times to remove free APAM and chitosan polymers. The formed APAM/chitosan membranes were then placed above hydrophilic silicon wafers. After freezing under −80 °C overnight, the membranes attached to silicon wafers were transferred to a vacuum freeze dryer (FD-1D-50, BIOCOOL) for 2–3 days. The freeze-dried samples were kept in an electronic dry cabinet before being cracked to expose the cross-sections for thickness measurement. We treated assemblies with a sputter coater (Bal-tec SCD 005) for conductive coating and observed the cross-sections under a scanning electron microscope (LEO 1530 FEG-SEM, Zeiss, Model S4800, Hitachi). SEM images of membrane cross-sections were collected and measured with the open-source software ImageJ.

### Thickness measurement of hydrated membranes with a confocal microscope

APAM solution was mixed with 0.005 wt% rhodamine 6 G (R6G, Mw = 479.01 g mol⁻¹, purchased from Aladdin, China). Chitosan solution was mixed with 0.005 wt% fluorescein sodium salt (Mw =376.27 g mol⁻¹, purchased from Aladdin, China). 50 μL APAM was placed above a thin cover slide, and around 30–50 μL chitosan solution was placed beside the APAM solution. The cover slide was placed above a 100x objective immersed in a drop of oil, so membranes on the interface between

APAM and chitosan solution were observed with the confocal microscope (Eclipse Ti2-E, Nikon). Confocal images of cross-sections of membranes were collected and measured with ImageJ.

## Selective transport across channel walls

Fluorescent dye solutions were prepared by dissolving fluorescein isothiocyanate-dextran 4000 (FITC-dex 4k, Mw = 4000 g mol⁻¹, purchased from Sigma-Aldrich, United States), fluorescein isothiocyanate-dextran 10,000 (FITC-dex 10k, Mw = 10,000 g mol⁻¹, purchased from Sigma-Aldrich, United States), fluorescein isothiocyanate-dextran 40,000 (FITC-dex 40k, Mw = 40,000 g mol⁻¹, purchased from Sigma-Aldrich, United States), and fluorescein isothiocyanate-dextran 70,000 (FITC-dextran 70k, Mw = 70,000 g mol⁻¹, purchased from Sigma-Aldrich, United States) in deionized water, respectively. 2 mm width channels were printed with 1 wt% chitosan printing ink (pH ≈ 6) and 2 wt% APAM printing matrix (pH ≈ 7), and the time for membrane assembly was 20 min. After removing the APAM matrix and the chitosan ink with deionized water, channels were perfused with fluorescent dye solutions at a constant flow rate (0.5 mL h⁻¹). During the perfusion, channels were immersed in water and observed under a fluorescence optical microscope. The collected fluorescence microscope images are analyzed with the software MATLAB to obtain the normalized fluorescence intensities.

## Localized trans-wall dye transport on a Y-shaped channel

The yellow dye and the red dye perfused in the channel were prepared by dissolving T/Thioflavin T (Mw = 318.86 g mol⁻¹, ≥75%, purchased from RYON, China) and rhodamine 6 G (R6G, Mw = 479.01 g mol⁻¹, purchased from Aladdin, China) in deionized water, respectively. The blue dye placed above the channel was prepared by dissolving methylene blue (MB, Mw = 319.85 g mol⁻¹, ≥82%, purchased from Sigma-Aldrich, United States) in deionized water. Before characterizing the real-time concentration of MB inside the channel, 2 mm-width channels were printed and perfused with deionized water at a flow rate of 3 mL h⁻¹. After placing a 5 μL MB droplet above the channel wall, we collected 100 μL downstream liquids from the channel outlet every 2 min. The distance between the channel outlet and the position for placing MB droplets was 1 cm. MB concentrations in the collected liquids were characterized with a Microplate Reader (SpectraMax iD3, Molecular Devices), as illustrated in Supplementary Fig. 20.

## Localized immobilization of enzymes on channel walls

For visualizing the immobilized enzymes on channel walls (Fig. 3), fluorescein isothiocyanate-labeled horseradish peroxidase (FITC-HRP, customized by Xi'an Ruixi Biological Technology Co. Ltd, China) and rhodamine B-labeled glucose oxidase (RhB-GOx, customized by Xi'an Ruixi Biological Technology Co. Ltd, China) were used. The enzymes used for glucose degradation in Fig. 5d were unlabeled HRP (Pierce Horseradish Peroxidase, 300 units mg⁻¹, purchased from Thermo Fisher Scientific, United States) and GOx (Glucose oxidase, 100 units mg⁻¹, purchased from Macklin, China). Enzymes were dissolved in deionized water and stored at −20 °C before usage. To immobilize positively charged horseradish peroxidase (HRP) on channel walls, HRP solutions were deposited close to the walls for 30 min, during which some enzymes adsorb onto the wall surface. The residual enzymes were washed away with deionized water. To immobilize negatively charged glucose oxidase (GOx) on channel walls, the wall surface is pre-coated with a chitosan layer. The chitosan layer was generated by placing chitosan solutions in contact with the channel walls for 30 min, followed by remove of the free chitosans by washing with deionized water. Then, GOx solutions were left in contact with the walls for 30 min for the adsorption of enzymes onto channel wall surface. To generate a GOx-HRP co-coated layer on walls in Fig. 3b(ii), the channel wall was pre-coated with HRP enzymes. The outer surface coated with positively charged HRP can attract negatively charged

GOx for GOx immobilization. In Fig. 3b, the left GOx-coated region was treated with 0.5 μL, 10 mg mL⁻¹ RhB-GOx solution; the middle GOx-coated region was treated with 0.5 μL, 4 mg mL⁻¹ RhB-GOx solution; the right GOx-coated region was coated with 1 μL, 1 mg mL⁻¹ RhB-GOx solution; the left HRP-coated region was treated with 0.5 μL, 1 mg mL⁻¹ FITC-HRP aqueous solution, and the right was coated with 1 μL, 4 mg mL⁻¹ FITC-HRP aqueous solution. The GOx-coated region and HRP-coated region in Fig. 5d are treated with 2 μL GOx aqueous solution (5 mg mL⁻¹) and 2 μL HRP aqueous solution (5 mg mL⁻¹), respectively.

## Characterization of glucose and starch concentrations inside channel

The solution mixture of starch and glucose was prepared by dissolving glucose (Dextrose, Mw = 180.16 g mol⁻¹, anhydrous, purchased from Sigma-Aldrich, United States) and starch (Starch from corn, pharmaceutical, purchased from Aladdin, China) in deionized water. The mixed solution contained 20 wt% starch and 20 wt% glucose. 2 μL starch-glucose solution was placed above a 2 mm-width channel perfused with deionized water inside (flow rate = 1.5 mL h⁻¹). The distance between the channel outlet and the position for placing the starch-glucose solution was 1 cm. 150 μL downstream liquids were collected from the channel outlet every 6 min. The glucose concentration in the collected liquids was analyzed with glucose assay kit reagents (Glucose Assay Kit with O-toluidine, purchased from Beyotime, China). The starch concentration in the collected liquids was analyzed with starch assay kit reagents (Starch Content Assay Kit, Sulfuric acid anthrone colorimetric method, purchased from Solarbio, China).

## Detection of $CO_2$ concentrations outside the channel

Setups for detecting $CO_2$ concentration surrounding the channel were presented in Supplementary Fig. 25. The $CO_2$ sensor (CM-0024 10,000 ppm $CO_2$ Sensor, CO2Meter) provided data on $CO_2$ concentration every 5 s.

## Localized trans-wall dye transport and enzyme immobilization

A VasFluidic channel was printed and exposed to the air, as presented in Fig. 4b. The solution of green fluorescent microspheres (Diameter = 1 μm, excitation peak = 488 nm, emission peak = 518 nm, surface group: -COOH, 10 mg mL⁻¹, purchased from Tianjin BaseLine Chromtech Research Center, China) was mixed with FITC-HRP solution (10 mg mL⁻¹) at a volume ratio of 1:1. The mixed solution was then placed above 4 downstream channels for 30 min before washing with water. Droplets of R6G and fluorescein sodium aqueous solutions were placed above 4 upstream channels when the channel was perfused with water inside. The channel was exposed under 405 nm light (Portable curing light source, 405 nm, 3 W, purchased from Engineering For Life, China) during data collection.

## Data availability

All data needed to evaluate the conclusions in the paper are presented in the paper and/or the Supplementary Information, and also from the corresponding author upon request. Source data are provided with this paper.

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

## Acknowledgements

We appreciate Dr. Yang Xiao for helping to repair the 3D printer. We are thankful to Mr. Qinyu Li, Dr. Xiufeng Li, and Dr. Xiaolai Li for guidance in measuring Young's modulus of membranes with AFM microscope. The authors are grateful to Ms. Qingchun Song and Ms. Yang Cao for providing enzymes or enzyme-related reagents. The authors appreciate Mr. Yuchao Wang for checking the Supplementary Notes. The authors thank Mr. Vyshnav for the help in preliminary experiments on membrane permeability. The authors thank Ms. Qingchun Song, Ms. Qianwen Chen, and Mr. Zhenyu Yang for their help in using confocal scanning microscopes. The authors thank Dr. Tiantian Kong from Shenzhen University and Dr. Yau Kei Chan from the University of Hong Kong for helpful discussions and suggestions. The authors also thank Mr. Feipeng Chen, Mr. Huanqing Cui, Dr. Shipei Zhu, and Dr. Sammer Ul Hassan for their advice on experiment design. The authors acknowledge the financial support provided by the National Natural Science Foundation of China (NSFC)/ Research Grants Council of Hong Kong (RGC) Joint Research Scheme (No. N_HKU718/19), General Research Fund (Nos. 17306820, 17306221, 17317322) from RGC, and Excellent Young Scientists Fund (Hong Kong and Macau) (21922816) from NSFC, which were all received by H.C.S. H.C.S. was funded in part by the Croucher Senior Research Fellowship from Croucher Foundation, and the Health@InnoHK programme from the Innovation and Technology Commission of the Hong Kong SAR government.

## Author contributions

H.C.S. and Y.P. supervised the project. Y.Y. proposed the idea, designed and performed the majority of the experiments, analyzed the data, and wrote the original draft. Y.Y. and Y.S. performed preliminary experiments in the initial stage of the project. Y.P., Y.Y., J.T., R.Z., W.G., C.L., and H.C.S.

participated in data analysis, experiment design, and manuscript revision. All authors have given approval to the final version of the paper.

## Competing interests

H.C.S. is a scientific advisor of EN Technology Limited and Micro-Diagnostics Limited in both of which he owns some equity, and also a managing director of the research center, namely Advanced Biomedical Instrumentation Center Limited. The works in the paper are however not directly related to the works of these entities. The remaining authors declare no competing interests.
