## [Peer Review File · Nature Communications]

REVIEWER COMMENTS

Reviewer #1 (Remarks to the Author):

The manuscript presents a method to pattern thin-walled channels in polystyrene cell culture-treated dishes, giving rise to microfluidics patterns with versatile geometries. The channels – characterized by specific permeability settings – are used to pattern molecules on their outside part, so reactions occurring in the tubes can be modulated, for example, in their kinetics; or to introduce new molecules to the reactions on-demand. The presented concept has novelty and possible wide applicability. Although the concept is interesting, a more systematic exploration of the versatility of the system would be important to elucidate and broaden its applicability, namely for a vast audience.

Only one experimental condition is explored throughout the manuscript for several proof-of-concept experiments. Although the range of experimental conditions suitable for printing at fixed pH values is presented in Supporting Information, as well as the range of tube thicknesses obtained with different experimental settings (varying time, concentrations), the effect of experimental conditions on the mechanical and permeability properties of the tubes is not described. These data are important to prove the versatility of the proposed system beyond mimicking blood vessels, providing further possibilities for their use as adjustable microfluidic devices.

The operating range of the developed devices is also poorly assessed. Specifically, the range of minimum to maximum possible flow rates in the vessels are not explored. It would be important to define ranges for tube bursting, or even for the possible detachment of the tube walls from the adherent dish. The maximum resolution of the technique also requires further elucidation – how thin can the channels be printed is an important aspect to be considered. Additionally, reporting the maximum size for the acceptable function and maintenance (without collapse) of the channels may also be valuable to understand the possible working parameters of the proposed systems.

The analogy to native tissues/vessels would be reinforced by the characterization of the mechanical properties of the formed tubes (e.g., by AFM).

Membrane thicknesses were measured after freeze-drying the membranes. It is likely that freezing of the samples, and the consequent formation of water crystals, and further dehydration may create artifacts on the properties of the membranes. Confocal microscopy of labelled hydrated samples would enable a more reliable assessment of membrane thickness in their real working conditions.

The description of the experimental procedure for enzyme immobilization requires more details, namely concerning which device was used to pattern the enzyme on the channels.

Reviewer #2 (Remarks to the Author):

Summary report:

This manuscript discusses the fabrication and transport of material in a 3D printed fluidic system. The authors present a new approach named “VasFluidics” that enables selective transport of materials across the printed frame and surrounding media. The fluidic channels have thin (less than one micrometer thickness), elastic, and semipermeable walls, mimicking the natural fluidic system. The manuscript presents a well-researched concept that addresses some challenges in synthetic fluidic systems. The idea is clearly communicated, figures and supporting materials are high quality, and experimental section seems to provide enough information to replicate the experiments. I have following specific comments.

Comments:

Results:

- 1) It is not clear how the authors quantified the self-assembly at the interface? This is the first and major step in establishing the introduced printing technique. The mechanical properties of the porous wall have not been provided. It is unclear whether one can potentially take the printed channel without the solid wall support.
- 2) Figure 1: has there been any measurement that shows the distribution of charges on the surface (zeta potential of the solid surface)?
- 3) The authors need to clarify what the self-adapt to pressure fluctuations means? What is the maximum normal and shear stress that the channel walls can tolerate?
- 4) Examining the cross-section of printed channels hints that the printed channels walls are not uniform: the bottom wall is made of chitosan attached to the petri dish while the top part is made of APAM- chitosan. Such a non-uniformity can potentially affect the upcoming applications in terms of composition distribution, reactions, etc. The authors need to clarify such effects and if possible, provide remedies.
- 5) The authors need to support the following argument “The flexibility of the membranes is presumably attributed to the long polymer chains in high-molecular chitosan and APAM.” Has there been any attempt to repeat these (similar) experiments with any short-chained polymers?
- 6) Have the printed channels self-healing characteristics?
- 7) How the change (extension) on the duration of effective drug concentration is possible in VasFluidic channels?
- 8) Is there any validation for the temporal programmability of the methylene blue (MB) in the channel?

9) What is the maximum volume of intake fluid in localized trans-wall introduction and extraction (Figure 1c)?

10) Data on the stability of the printed channels (whether kept in the polymer solution or dried in air) is required.

11) How were the immobilized regions of GOx and HRP created within the midstream channel?

12) Perfusion system: what is the composition of the gelation agent used to seal the needle to the channel?

Point-by-point response letter to reviewer's comments for manuscript:

Manuscript Number: NCOMMS-23-14375

Vascular network-inspired fluidic system (VasFluidics) with spatially functionalizable membranous walls

Reviewer #1

General Comment: The manuscript presents a method to pattern thin-walled channels in polystyrene cell culture-treated dishes, giving rise to microfluidics patterns with versatile geometries. The channels – characterized by specific permeability settings – are used to pattern molecules on their outside part, so reactions occurring in the tubes can be modulated, for example, in their kinetics; or to introduce new molecules to the reactions on-demand. The presented concept has novelty and possible wide applicability. Although the concept is interesting, a more systematic exploration of the versatility of the system would be important to elucidate and broaden its applicability, namely for a vast audience.

Reply: We are very grateful to the reviewer for appreciating the concept of our work, as well as the helpful suggestions to improve our manuscript. As suggested, we did more systematic explorations on our fluidic system to elucidate the versatility. Please find below our point-by-point responses to the questions:

Comment 1: Only one experimental condition is explored throughout the manuscript for several proof-of-concept experiments. Although the range of experimental conditions suitable for printing at fixed pH values is presented in Supporting Information, as well as the range of tube thicknesses obtained with different experimental settings (varying time, concentrations), the effect of experimental conditions on the mechanical and permeability properties of the tubes is not described. These data are important to prove the versatility of the proposed system beyond mimicking blood vessels, providing further possibilities for their use as adjustable microfluidic devices.

Reply: We thank the reviewer for this excellent suggestion. As suggested, we measured the Young's modulus of membranous channel walls under saline, acid or alkaline conditions; indeed, the Young's modulus decreases under these conditions (Supplementary Fig. 7(b)). The Young's modulus is measured with atomic force microscope, as suggested in Comment 4. We also noted that the permeability of channel walls increases under saline and acid conditions, and decreases at alkaline conditions: More trans-wall FITC-Dex10 is observed at saline and acid conditions, comparing to that in neutral environment without saline. The trans-wall transport of

FITC-Dex10k is not observed as pH=11, although FITC-Dex10k can cross the channel wall under neutral pH. (Supplementary Fig. 16). We have added the additional figures and descriptions based on the newly conducted experiments to the revised manuscript.

Supplementary Figure 7(b). Young's modulus of membranous channel walls under different environments. Membranes are immersed in different solutions during the measurement with atomic force microscope.

Supplementary Figure 16. Fluorescence microscope images showing the trans-wall transport of FITC-Dex10k ($R_h \approx 1.9$ nm) under hypersaline, acidic or alkaline environment. The permeability of channel walls increases under saline and acid conditions, as more trans-wall FITC-Dex10 is observed within 60 min. The permeability of channel walls decreases with increasing pH, and FITC-Dex cannot cross the channel wall when increasing the pH to 11. The channel is infused with FITC-Dex solutions at a flow rate of 0.5 mL/h. Channel walls under view are exposed to 200 μ L deionized water. The normalized fluorescence intensities in images are analyzed with MATLAB. The scale bar is 1 mm.

Comment 2: The operating range of the developed devices is also poorly assessed. Specifically, the range of minimum to maximum possible flow rates in the vessels are not explored. It would be important to define ranges for tube bursting, or even for the possible detachment of the tube walls from the adherent dish.

Reply: Thank you for pointing out the important point regarding the operating flow rate of channels. To address this, we conducted new experiments where we infused liquids at different flow rates to determine allowable flow rates in channels of different size (Supplementary Fig. 14(c)). The membranous channel walls will detach from the substrate when the flow rate exceeds the allowable flow rates. The following data has been added to the revised manuscript.

* Maximum flow rates for our syringe pump is around 1680 mL h⁻¹, and flow rates higher than 1680 mL h⁻¹ is not applied here.

Supplementary Figure 14. Liquid perfusion of straight channels with different widths. 0.001-0.005 wt% rhodamine 6G aqueous solution is used as perfusion liquids for clear visualization. Channel outlets are exposed to air for smooth flow of internal liquids. The channel widths are measured before the printing matrix removal, corresponding to w shown in Supplementary Fig.10. (a) Liquid perfusion into a channel with width of 610 μm. The channel wall detach from the substrate as the flow rate reaches to 8 mL/h. The scale bar is 5 mm. (b) Liquid perfusion into a channel with width larger than 1 cm. The liquid flow rate is larger than 20 mL/h. The scale bars is 5 mm. (c) Allowable flow rates for 50-60 mm long straight channels with different widths. Channels are perfused under certain flow rates for at least 1 min to observe if liquid will leak out of the channel, or the channel walls will detach from the substrate.

Comment 3: The maximum resolution of the technique also requires further elucidation – how thin can the channels be printed is an important aspect to be considered. Additionally, reporting the maximum size for the acceptable function and maintenance (without collapse) of the channels may also be valuable to understand the possible working parameters of the proposed systems.

Reply: The minimum channel we obtained in the work has a width around 500 μm , and the maximum channel has a width around 2 cm (Supplementary Fig. 9(c) and (d)). Liquid infusion into micron-sized channel and centimeter-sized channel is presented in Figure R1.

The fabrication of larger channel is possible, but is not realized with our setups with limitations. The cross-sectional area (a) is highly relevant to the printing speed (P) and the printing ink flow rate (F), as illustrated in Supplementary Note 2, Supplementary Fig. 9 (a) and (b). Theoretically, the channel with larger cross-sectional area (a) can be obtained by decreasing the printing speed and /or increasing the ink flow rate. However, the printing speed cannot be decreased further with our current 3D printer; the ink flow rate cannot be further increased, since our current pump cannot provide a higher pressure to extrude the viscous chitosan ink via the printing nozzle.

Essentially, the printing of smaller channel is restricted by the size of our printing nozzle (inner diameter = 520-620 μm). However, the principles should be general and applicable to smaller channels. During the 3D printing of the channel, the nozzle deposits ink onto the substrate, similar to a pen writing on paper. To print thinner channels, nozzles with smaller tips are required; however, the focus of this work is on the general principle of the VasFluidic system. The printing of smaller channels, and their applications are indeed interesting in its own right, and deserve a separate further study, which we are pursuing in a separate project. To address this important comments, we have added some descriptions and analyses regarding the relationship between the size of the printed channels and the printing nozzles, as well as a corresponding figure to the revised Supplementary Information.

Supplementary Figure 9. (a) Illustration for printing. (b) Relationship between printing ink flow rate F ($\mu\text{L}/\text{s}$), print speed P (mm/s), and the cross-sectional area a (mm^2) of the printed channel. (c) Confocal laser scanning microscope images showing cross sections of channels printed with different printing parameters. The printing ink is pre-mixed with 0.02 wt% FITC-Chitosan for visualization. Images are collected before removing the printing ink and matrix. Boundaries of the cross-sections are highlighted with red dotted lines. The scale bar is 500 μm . (d) Photographs showing cross sections of channels with larger sizes. Boundaries of the cross-sections are highlighted with red dotted lines. The scale bar is 5 mm.

Figure R1. Liquid perfusion into (a) a channel with width of 610 μm and (b) a channel with a width larger than 1 cm. Scale bars are 5 mm.

Supplementary Note 2: Controllable size of printed channels

The size of printed channels is determined by the channel length and the area of cross-section perpendicular to the length. The channel length is decided by the printing distance D (unit: mm), as shown in Supplementary Fig. 8(a). The area of the cross-section perpendicular to the length is defined as a (mm^2), which can be adjusted by changing the printing speed P (mm/s) and the flow rate of printing ink (F , $\mu\text{L/s}$) (Supplementary Fig. 8(a)). Specifically, printing speed P refers to the moving speed of the print nozzle, determined by the print distance (D , unit: mm) of the nozzle per unit of printing time (T , unit: s) (Supplementary Fig. 8(a)):

$$P = D/T \quad (1)$$

Ink flow rate F determines the ink volume (unit: mm^3) being extruded from the print nozzle per unit of printing time T , which relates to the volume of the printed channel:

$$F \cdot T = D \cdot a \quad (2)$$

Equation 1 and Equation 2 figure out the theoretical relationship between a , F , and P :

$$a = F/P \quad (3)$$

By measuring channels with different cross-sectional areas (Supplementary Fig. 8(c)), we find the measured a matches well with Equation 3 (Supplementary Fig. 8(b)). Therefore, a of printed channels is predictable with known F and P (Equation 3).

Morphological parameters of the cross-sections can be further estimated with known values of a . We approximated morphologies of channels' cross-sections as parts of a circle, as shown in Supplementary Fig. 6(a). The height (h), the width (w), the length of the membrane part (m), and the length that channel attached to the substrate (s) were measured by analyzing confocal laser scanning images of channels' cross-sections in Supplementary Fig. 8(c). Since parameters of lengths (m , w , s , h) should have approximate square relation with the area (a), we approximated the relation between h (or w , m , s) and a using the following Equation 4, which fits the measured data well (Supplementary Fig. 9(b)):

$$h \text{ (or } w, m, s) = k_h \sqrt{a}, \text{ where } k_h \text{ (or } k_w, k_m, k_s) \text{ is a constant} \quad (4)$$

In this equation, w equals s with a constant value of 520-620 μm when $a \leq 0.1 \text{ mm}^2$. The constant value may relate to the inner diameter of the print nozzle, which is also 520-620 μm . Since the value of w can be directly measured under an optical microscope, a , h , and m can also be estimated using measured w when w is larger than 620 μm , as indicated in Equation 4.

Moreover, we found volumes of channels expand with the prolonged assembly time of membranes (T_{ma}), which can lead to estimation errors in Equation 3 and Equation 4. The change ratios of a, h, m, w, s are present in Supplementary Fig. 10. The volumetric expansion may result from the unbalanced osmotic pressure between the printing ink and matrix.

In our present work, channels we obtained have a widths (w) of around 500 μm-2 cm. The printing of smaller channels is restricted by the size of our printing nozzle; larger channels are not obtained due to the limitations of our setups. For instance, the printing speed cannot be decreased further with our current 3D printer, and our current pump cannot provide a higher pressure to extrude the viscous printing ink at a faster ink flow rate. Hence, although not realized by our setups yet, the fabrication of larger or smaller sized channels is possible with modified 3D printing setups.

Supplementary Figure 10. Morphological parameters of channels' cross-sections. (a) Morphologies of cross-sections are approximated as parts of a circle with a radius of r . The cross-sectional area, the height, the maximum width, the length of the APAM/chitosan membrane part, and the length that channel attached to the substrate are defined as a , h , w , m and s , respectively. (b) Relation between m (or w , s , h) and a . The morphological parameters are measured with confocal laser scanning microscope images in Figure S9(c). The dots are measured data of m (blue), w (green), s (red), and h (orange), respectively. The solid lines are corresponding fit curves, and the functions of the fit curves are listed in Equation 4 in Note S2. The measured data fit well with the fit curves, and thus m (or w , s , h) can be estimated with known values of a :

$$m = 2.687 \sqrt{a}, w = 1.399 \sqrt{a}, s = 1.132 \sqrt{a}, h = 0.851 \sqrt{a}.$$

Comment 4: The analogy to native tissues/vessels would be reinforced by the characterization of the mechanical properties of the formed tubes (e.g., by AFM).

Reply: We appreciate the reviewer for providing us with the possible characterization method. Using an AFM, Young's modulus of the channel walls is 1-9 kPa (Supplementary Fig. 7(a)), which is of the same order of magnitude of some soft tissues, such as skin, spleen, and pancreas (*Int. J. Mol. Sci.* **16**, 15997-16016 (2015)).

Supplementary Figure 7(a). Young's modulus of membranous channel walls with different thicknesses. The membranes are immersed under water to maintain hydrated during the measurement with atomic force microscope.

Comment 5: Membrane thicknesses were measured after freeze-drying the membranes. It is likely that freezing of the samples, and the consequent formation of water crystals, and further dehydration may create artifacts on the properties of the membranes. Confocal microscopy of labelled hydrated samples would enable a more reliable assessment of membrane thickness in their real working conditions.

Reply: In response to the reviewer’s helpful suggestion, we have measured the membrane thickness with confocal microscope. The measurement results of both hydrated and freeze-dried samples have been added to the revised manuscript with the caption, “*Measurements of hydrated membrane walls may more closely represent their thicknesses as the channels are infused with liquids during operation*” (Supplementary Fig. 6(a) and (b)).

Supplementary Figure 6. Thickness of hydrated and freeze-dried membrane walls. Measurements of hydrated membrane walls may more closely represent their thicknesses as the channels are infused with liquids during operation. The thickness of hydrated membranes are measured with confocal microscope images. The chitosan solution is mixed with 0.005 wt% fluorescein sodium, and the APAM solution is mixed with 0.005 wt% rhodamine 6G for visualizing membrane cross-sections under confocal microscope. The thickness of freeze-dried membranes are measured with scanning electron microscope images. (a) Thicknesses of membranous walls with various membrane assembly time. 1wt% chitosan solution (pH≈6) is used to react with APAM solution (pH≈7) for membrane assembly. (b) Changing thickness of membranes by changing APAM concentrations in APAM printing matrix. 1wt% chitosan solution (pH≈6) is used to react with APAM solution (pH≈7) for membrane assembly, and the duration for membrane assembly is 20 min.

Comment 6: The description of the experimental procedure for enzyme immobilization requires more details, namely concerning which device was used to pattern the enzyme on the channels.

Reply: We thank the reviewer for the important comment. In response, we have added more details in the METHODS part of the revised manuscript: “Enzymes were dissolved in deionized water and stored at -20 °C before usage. To immobilize positively charged horseradish peroxidase (HRP) on channel walls, HRP solutions were deposited close to the walls for 30 min, during which some enzymes adsorb onto the wall surface. The residual enzymes were washed away with deionized water. To immobilize negatively charged glucose oxidase (GOx) on channel walls, the wall surface is pre-coated with a chitosan layer. The chitosan layer was generated by placing chitosan solutions in contact with the channel walls for 30 minutes, followed by remove of the free chitosans by washing with deionized water. Then, GOx solutions were left in contact with the walls for 30 min for adsorption of enzymes onto channel wall surface. To generate a GOx-HRP co-coated layer on walls in Fig.3b (ii), the channel wall was pre-coated with HRP enzymes. The outer surface coated with positively charged HRP can attract negatively charged GOx for GOx immobilization. In Fig. 3(b), the left GOx-coated region was treated with 0.5 μ L, 10 mg/mL RhB-GOx solution; the middle GOx-coated region was treated with 0.5 μ L, 4mg/mL RhB-GOx solution; the right GOx-coated region was coated with 1 μ L, 1mg/mL RhB-GOx solution; the left HRP-coated region was treated with 0.5 μ L, 1 mg/mL FITC-HRP aqueous solution, and the right was coated with 1 μ L, 4 mg/mL FITC-HRP aqueous solution. The GOx-coated region and HRP-coated region in Fig. 5(d) are treated with 2 μ L GOx aqueous solution (5 mg/mL) and 2 μ L HRP aqueous solution (5 mg/mL), respectively”.

Reviewer #2

Summary report: This manuscript discusses the fabrication and transport of material in a 3D printed fluidic system. The authors present a new approach named “VasFluidics” that enables selective transport of materials across the printed frame and surrounding media. The fluidic channels have thin (less than one micrometer thickness), elastic, and semipermeable walls, mimicking the natural fluidic system. The manuscript presents a well-researched concept that addresses some challenges in synthetic fluidic systems. The idea is clearly communicated, figures and supporting materials are high quality, and experimental section seems to provide enough information to replicate the experiments. I have following specific comments.

Reply: We thank the reviewer for the positive assessment and in-depth reading of our manuscript. Please find below our point-by-point responses to the questions raised:

Comment 1: It is not clear how the authors quantified the self-assembly at the interface? This is the first and major step in establishing the introduced printing technique. The mechanical properties of the porous wall have not been provided. It is unclear whether one can potentially take the printed channel without the solid wall support.

Reply: This is indeed an important point; we have conducted more quantifications on the self-assembly, including Young’s modulus of the self-assemblies (Supplementary Fig. 7), the thickness of assemblies (Supplementary Fig. 6), the changes of permeability (Supplementary Fig. 16) and Young’s modulus (Supplementary Fig. 7(b)) of membranes under saline, acid or alkaline conditions. Besides, the self-assembly process at the liquid interface is visualized under confocal microscope (Supplementary Fig. 4). Using these new data, the measurement of Young’s modulus reflects the mechanical property of walls in elasticity. All these additional data and figures have been added to the supplementary information of the revised manuscript.

As to “whether one can potentially take the printed channel without the solid wall support”, our answer is no. Without the solid wall support, the soft channel walls cannot retain the designed architecture during the removal of printing matrix. More relevant details are presented here:

Why the printing matrix should be removed after channel fabrication: “*The matrix removal provides access to localize the adjustment of fluids inside, for instance, by confining solutions outside a specific channel for local reagent delivery, or by immobilizing a local channel wall with enzymes to alter flow-through molecules.*”

Why channels without support can lose the designed architecture during the removal: “*The walls can be soft, semipermeable, and ultrathin (less than 1 μm)²² to mimic functions of biological soft tissue^{16, 18, 19, 29, 30} Nevertheless, when isolated from the matrix, the self-assembled channels are too soft to retain 3D-printed architectures at the designated location³⁰. Hence, reported channels are typically embedded within*

the matrix¹⁶⁻²⁸, only allowing overall homogeneous molecular exchange between the matrix and liquids inside^{16-20, 25}.”

Why we immobilize the channel onto the solid substrate: “Facilitated by embedded 3D printing, flexible, thin (1-2 μm), and semipermeable walls are immobilized to a solid substrate, similar to soft tissues supported by bone tissues, to avoid damaging or translocating the printed configurations upon matrix removal.”

Supplementary Figure 7. (a) Young's modulus of membranous channel walls with different thicknesses. The membranes are immersed under water to stay hydrated during the measurement with atomic force microscope. (b) Young's modulus of membranous channel walls under different environments. Membranes are immersed in different solutions during the measurement with atomic force microscope.

Supplementary Figure 6. Thickness of hydrated and freeze-dried membrane walls. Measurements of hydrated membrane walls may more closely represent their thicknesses as the channels are infused with liquids during operation. The thickness of hydrated membranes is measured with confocal microscope images. The chitosan solution is mixed with 0.005 wt% fluorescein sodium, and the APAM solution is mixed with 0.005 wt% rhodamine 6G for visualizing membrane cross-sections under confocal microscope. The thickness of freeze-dried membranes is measured with scanning electron microscope images. (a) Thicknesses of membranous walls with various membrane assembly time. 1 wt% chitosan solution ($\text{pH} \approx 6$) is used to react with APAM solution ($\text{pH} \approx 7$) for membrane assembly. (b) Changing thickness of membranes by changing APAM concentrations in APAM printing matrix. 1 wt% chitosan solution ($\text{pH} \approx 6$) is used to react with APAM solution ($\text{pH} \approx 7$) for membrane assembly, and the duration for membrane assembly is 20 min. (c) Changing thickness of membranes by changing the pH of chitosan printing ink. 1 wt% chitosan solutions with different pH are used to react with APAM solution (1 wt%, $\text{pH} \approx 7$) for membrane assembly, and the duration for membrane assembly is 20 min. Thickness of freeze-dried samples are measured based on the SEM images, and the thickness of hydrated samples are not measured. Hydrated samples are not clearly observed under confocal microscope due to the decreased fluorescence intensity of sodium fluorescein under low pH.

Supplementary Figure 16. Fluorescence microscope images showing the trans-wall transport of FITC-Dex10k ($R_h \approx 1.9$ nm) under hypersaline, acidic or alkaline environment. The permeability of channel walls increases under saline and acidic conditions, as more trans-wall FITC-Dex10 is observed within 60 min. The permeability of channel walls decreases with increasing pH, and FITC-Dex cannot cross the channel wall when increasing the pH to 11. The channel is infused with FITC-Dex solutions at a flow rate of 0.5 mL/h. Channel walls under view are exposed to 200 μ L deionized water. The normalized fluorescence intensities in images are analyzed with MATLAB. The scale bar is 1 mm.

Supplementary Figure 4. Aggregation of polymers on the interface between APAM and chitosan solutions. 1 wt% chitosan solution is pre-mixed with 0.005 wt% fluorescein sodium, and 1 wt% APAM solution is pre-mixed with 0.005 wt% rhodamine 6G for visualization under confocal microscope. The scale bar is 10 μ m.

Comment 2: Figure 1: has there been any measurement that shows the distribution of charges on the surface (zeta potential of the solid surface)?

Reply: We thank the reviewer for reminding us to measure the charges on the surface. Both the tissue culture-treated dishes and non-treated dishes can be utilized as substrate, which have surface carrying negative charges. The surface charge density of tissue culture-treated dishes is $-1.81 \pm 0.53 \mu\text{C}/\text{cm}^2$, and the charge density of non-treated ones is $-1.50 \pm 0.36 \mu\text{C}/\text{cm}^2$. Details on the measurement are now included in the METHODS part of our manuscript: “*Surface charge density measurement of Petri dishes: The Petri dishes were cut by a clean scissor into small pieces with areas of approximately 10 cm^2 . The exact area was measured after the surface charge density measurement. The small pieces were transferred by a clean tweezer into a Faraday cup. The Faraday cup was connected with a programmable electrometer (6514, Keithley Instruments Model) under the charge measurement model. For measurements of the same petri dish type, pieces were collected from 3 petri dishes, and each piece was measured 3 times. The surface charge densities were calculated by dividing the measured charge value by the measured area of samples.*”

Comment 3: The authors need to clarify what the self-adapt to pressure fluctuations means? What is the maximum normal and shear stress that the channel walls can tolerate?

Reply: We thank the reviewer for pointing out the potentially confusing expression, “self-adapt to pressure fluctuations”, which has now been deleted. In addition, we have revised the corresponding part to “*Due to the elasticity of the walls, the channels can be inflated or collapsed to alter intracavity volume in response to the changes in liquid volume inside...*”.

Moreover, in addition to providing the maximum shear stress that the channel walls can tolerate, we have conducted additional experiments where channels are infused at different flow rates to determine allowable flow rates for channels of different size (Supplementary Fig. 14). These have now been added to the revised manuscript.

* Maximum flow rates for our syringe pump is around 1680 mL h⁻¹, and flow rates higher than 1680 mL h⁻¹ is not applied here.

Supplementary Figure 14. Liquid perfusion of straight channels with different widths. 0.001-0.005 wt% rhodamine 6G aqueous solution is used as perfusion liquids for clear visualization. Channel outlets are exposed to air for smooth flow of internal liquids. The channel widths are measured before the printing matrix removal, corresponding to w shown in Supplementary Fig.10. (a) Liquid perfusion into a channel with width of 610 μm. The channel walls detach from the substrate as the flow rate reaches to 8 mL/h. The scale bar is 5 mm. (b) Liquid perfusion into a channel with width larger than 1 cm. The liquid flow rate is larger than 20 mL/h. The scale bars is 5 mm. (c) Allowable flow rates for 50-60 mm long straight channels with different widths. Channels are perfused under certain flow rates for at least 1 min to observe if liquid will leak out of the channel, or the channel walls will detach from the substrate.

Comment 4: Examining the cross-section of printed channels hints that the printed channels walls are not uniform: the bottom wall is made of chitosan attached to the petri dish while the top part is made of APAM- chitosan. Such a non-uniformity can potentially affect the upcoming applications in terms of composition distribution, reactions, etc. The authors need to clarify such effects and if possible, provide remedies.

Reply: Thank you for your constructive comment. Non-uniform channel wall is not uncommon among many reported fluidic systems, including open fluidic system with one side exposed to the air (*Open-space microfluidics: concepts, implementations, applications[M]. John Wiley & Sons, 2018*), and some PDMS-based fluidic systems bonding with other materials as substrates (*Biosensors, 2021, 11(8): 292*). The influence of such non-uniform channel walls on the fluid composition distribution needs further investigation. Despite the non-uniformity of our channel walls, the influence on composition distribution could be reduced by the extra-coating on the inner surface of the channel walls. Since the inner surface assembled by chitosans is positively charged, the inner surface can be coated with multiple layers via layer-by-layer assembly under electrostatic forces as reported (*Advanced Materials, 2022, 34(5): 2105386*). To address this important point, a brief discussion of the feature of non-uniform channel wall and the corresponding references have been added to the revised manuscript.

Comment 5: The authors need to support the following argument “The flexibility of the membranes is presumably attributed to the long polymer chains in high-molecular chitosan and APAM.” Has there been any attempt to repeat these (similar) experiments with any short-chained polymers?

Reply: We thank the reviewer for this helpful comment. We have attempted to utilize chitosan polymers with shorter chains (Chitosan, < 200 mPa.s, purchased from Aladdin, China). When the short-chained chitosans are used, the interfacial self-assemblies are not robust enough to lock the chitosans inside. Some chitosans leak out from the inside, spreading around and generate formless complexes with APAM outside, as shown in Figure R2. In our revised manuscript, we have toned down the original claim on the role of the chain length of the polymers, and added this additional information using short-chained chitosan to the “*printing ink and printing matrix preparation*” section of *Methods* part.

Figure R2. Printed chambers with formless APAM/chitosan walls by using short-chained chitosans. Some short-chained chitosans leak out, generating formless coacervates with APAM around, which are pointed out by red arrows. The chamber is produced by depositing the chitosan solution (1 wt% , pH \approx 6) is within the APAM matrix, and the coacervate assembly time is 30 min. The scale bar is 1 mm.

Comment 6: Have the printed channels self-healing characteristics?

Reply: The channel walls can heal themselves before removing the printing ink and matrix, as shown in the additional data in Supplementary Fig.8. Once the channel walls are punctured, chitosan ink inside the chamber can react with APAM matrix outside to generate new chitosan/APAM assemblies to “heal” the walls. However, such new assemblies cannot form if printing ink and matrix are removed, and thus channel walls cannot self-heal after the ink and matrix removal (Supplementary Fig.15). We have highlighted the important information in the revised manuscript.

Supplementary Figure 8. Self-healing characteristics of membranous walls of printed chambers. (a) Membranes self-heal after being punctured by a sharp needle. Once the membranous complexes are punctured, chitosan ink inside the chamber can react with APAM matrix outside to generate new chitosan/APAM assemblies. The coacervation time is 30 minutes for the chamber in the image labeled 0 s. The scale bar is 1 mm. (b) New branches can be added to an established chamber, resulting from the self-healing properties of the membranous walls. The membrane assembly time is 30 minutes for the chamber in the image labeled 0 s. The scale bar is 1 mm. (c) Due to the self-healing properties of membranous walls, it is feasible to build new bridges between two separate chambers or to cut established bridges. Upon injecting dye liquids into the left-side channel, liquids can pass through the new bridge without leaking out, in comparison to the case when the channel has been truncated and liquid cannot flow through. The scale bar is 3 mm.

Supplementary Figure 15. The channel walls cannot self-heal after removing the printing ink and matrix. (a) A channel filled with water inside and exposed to the air. The channel wall is punctured with a sharp needle. The channel does not self-heal within 1 hour, which is confirmed by (b) perfusing the channel with dye solutions, during which the dye leaks out from the puncture. The scale bar is 5 mm.

Comment 7: How the change (extension) on the duration of effective drug concentration is possible in VasFluidic channels?

Reply: In our manuscript, we mentioned “*the duration of effective drug concentration can be extended by repeatedly introducing the drug inside.*” To demonstrate this, we place a 5 μL , 0.2 wt% methylene blue (MB) solution above the channel, and after 30 minutes, the downstream MB concentration is below 0.001 wt‰, as shown in Fig.2(d). However, by repeatedly replenishing and depositing MB solutions onto the channel every 20 min, the downstream MB concentration can stay above 0.001 wt‰ for 60 min (Supplementary Fig. 19). Similarly, the drug concentration in channel can also be maintained above a certain level by repeatedly introducing the drug via trans-wall transport. To elaborate on this, these additional figures and descriptions have been added to the revised manuscript.

Fig.2d Temporal regulation of MB concentration inside the channel via localized trans-wall MB transport. The width of the channel is 2 cm. The assembly time of membranes is 15 min. The channel is perfused with deionized water at a flow rate of 3 mL/h. After placing a 5 μL MB droplet beside the channel, 1 cm downstream liquids inside are collected every 2 min to analyze the MB concentration inside channel. MB concentrations with different dynamics are presented when (i) different concentrated-MB droplets are placed beside the channel, or (ii) channels with different wall thicknesses are applied.

Supplementary Figure 19. MB concentration inside channel can be maintained above 0.001 wt % by introducing 5 μ L, 0.2 wt% MB solutions every 20 min. Similarly, the drug concentration in channel can also be maintained above a certain level by repeatedly introducing the drug via trans-wall transport.

Comment 8: Is there any validation for the temporal programmability of the methylene blue (MB) in the channel?

Reply: The temporal programmability has been presented in Fig.2d of the manuscript. For instance, after introducing a 0.1 wt% MB droplet, the downstream MB becomes undetectable after 30 min; in contrast, if a 0.4 wt% MB droplet is used instead, it takes 50 min for the MB to be undetectable (Fig.2d(i)). Hence, the availability of the MB can be programmed by varying the MB concentration of the liquid droplet introduced. Apart from the programmability in duration, the time for the peak concentration of MB can also be controlled. For example, the peak time of the downstream MB concentration can be postponed from 4 min to 8 min by increasing channel wall thickness from 1.2 μm to 1.4 μm (Fig.2d(ii)). These additional data confirm the programmability offered by VasFluidics systems and have been added to the revised manuscript.

Fig.2d Temporal regulation of MB concentration inside the channel via localized trans-wall MB transport. The width of the channel is 2 cm. The assembly time of membranes is 15 min. The channel is perfused with deionized water at a flow rate of 3 mL/h. After placing a 5 μL MB droplet beside the channel, liquids are collected 1 cm downstream from the droplet position every 2 min to analyze the MB concentration inside channel. MB concentrations with different dynamics are presented when (i) different concentrated-MB droplets are placed beside the channel, or (ii) channels with different wall thicknesses are applied.

Comment 9: What is the maximum volume of intake fluid in localized trans-wall introduction and extraction (Figure 2c)?

Reply: The maximum volume varies with the size and geometry of the fabricated channels; in our experiment, up to 8 mL liquids can be placed for the geometry used (Supplementary Figure 17). We have added this useful piece of information to the revised manuscript.

Fig.2c Spatiotemporal regulation of fluid components via the localized trans-wall introduction and extraction. (i) A Y-shaped channel is perfused with rhodamine 6G (red) and T/Thioflavin T (yellow) aqueous solutions. (ii) Dyes at random positions and times can be extracted by depositing one deionized water droplet next to the channel. (iii) Methylene blue (MB, a blue dye) can be introduced into the channel at random positions and times by placing an MB droplet next to the channel. The scale bar is 1 cm.

Supplementary Figure 17. Localized trans-wall introduction by depositing solutions on or next to the channel. The volume of the placed solution can be increased from microliters to milliliters. Methylene blue (MB) aqueous solution is used for easy visualization. Pipette tip is utilized to deposit the dye solution. The channel is infused with water inside at a flow rate of 2 mL/h during the deposition of MB solutions. The channel has been infused with water for at least 30 min before we place solutions next to the channel. The scale bar is 1 cm.

Comment 10: Data on the stability of the printed channels (whether kept in the polymer solution or dried in air) is required.

Reply: As suggested, we have conducted additional experiments on the stability of the printed channels. Based on our observations, the channel cannot be left in air for complete drying, otherwise the channel will collapse with membrane walls completely adhere to the substrate, as shown in Supplementary Fig. 24(a). The dried channels cannot be restored to the original structure after rehydration (Supplementary Fig. 24(b)).

However, the channel can be stored for at least 7 days if the membranous walls are kept hydrated. After the 7-day storage under humid environment, the channel maintains the printed shape and allows liquid infusion (Supplementary Fig. 25). We have added these stability data to the revised manuscript, as suggested by the reviewer.

Supplementary Figure 24. (a) Liquid cannot be injected in the dried channel. (b) The dried channel cannot be restored to the original structure even after re-hydration. Scale bars are 1 cm.

Supplementary Figure 25. Printed channel can be stored for 7 days, after which the channel still maintains the original structure and allows the infusion of liquids inside. During the storage in our experiments, APAM matrix outside the channel is removed and chitosan ink inside the channel is retained. Water is placed near the channel to increase the surrounding humidity, and thus the channel wall can stay hydrated. The channel is then stored in a sealed container and placed under 4°C. The scale bar is 1 cm.

Comment 11: How were the immobilized regions of GOx and HRP created within the midstream channel?

Reply: The immobilization of GOx and HRP in Fig.5 follows that in Fig. 3, where fluorescent versions of GOx and HRP, namely RhB-GOx and FITC-HRP are used. To enhance the readability, we have also added more details about enzyme immobilization in the METHODS part of the revised manuscript: “*Localized immobilization of enzymes on channel walls: For visualizing the immobilized enzymes on channel walls (Fig. 3), fluorescein isothiocyanate-labeled horseradish peroxidase (FITC-HRP, customized by Xi'an Ruixi Biological Technology Co. Ltd, China) and rhodamine B-labeled glucose oxidase (RhB-GOx, customized by Xi'an Ruixi Biological Technology Co. Ltd, China) were used. The enzymes used for glucose degradation in Fig. 5(d) were unlabeled HRP (Pierce Horseradish Peroxidase, 300 units/mg, purchased from Thermo Fisher Scientific, United States) and GOx (Glucose oxidase, 100 units/mg, purchased from Macklin, China). Enzymes were dissolved in deionized water and stored at -20 °C before usage. To immobilize positively charged horseradish peroxidase (HRP) on channel walls, HRP solutions were deposited close to the walls for 30 min, during which some enzymes adsorb onto the wall surface. The residual enzymes were washed away with deionized water. To immobilize negatively charged glucose oxidase (GOx) on channel walls, the wall surface is pre-coated with a chitosan layer. The chitosan layer was generated by placing chitosan solutions in contact with the channel walls for 30 minutes, followed by remove of the free chitosans by washing with deionized water. Then, GOx solutions were left in contact with the walls for 30 min for adsorption of enzymes onto channel wall surface. To generate a GOx-HRP co-coated layer on walls in Fig.3b (ii), the channel wall was pre-coated with HRP enzymes. The outer surface coated with positively charged HRP can attract negatively charged GOx for GOx immobilization. In Fig. 3(b), the left GOx-coated region was treated with 0.5 μ L, 10 mg/mL RhB-GOx solution; the middle GOx-coated region was treated with 0.5 μ L, 4mg/mL RhB-GOx solution; the right GOx-coated region was coated with 1 μ L, 1mg/mL RhB-GOx solution; the left HRP-coated region was treated with 0.5 μ L, 1 mg/mL FITC-HRP aqueous solution, and the right was coated with 1 μ L, 4 mg/mL FITC-HRP aqueous solution. The GOx-coated region and HRP-coated region in Fig. 5(d) are treated with 2 μ L GOx aqueous solution (5 mg/mL) and 2 μ L HRP aqueous solution (5 mg/mL), respectively.”*

Fig. 3 Spatial immobilization of functional materials on channel walls. a (i, ii) Schematic showing the localized immobilization of functional materials on channel walls by attaching solutions to the channel. The large-sized solutes (radius or $R_h \geq 3.0$ nm) carrying electrostatic charges cannot penetrate membrane walls but can be electrostatically immobilized on the membranes. (iii, iv) Schematics showing the immobilization of negatively or positively charged solutes on channel walls; Confocal laser scanning microscope images showing channel walls immobilized with the positively charged FITC-labeled horseradish peroxidase (FITC-HRP, green) or negatively charged RhB-labeled glucose oxidase (RhB-GOx, red). Scale bars are $200 \mu\text{m}$. b (i) Schematic and (ii) fluorescence microscope images showing the spatially arranged immobilization of the two enzymes on the channel walls. RhB-GOx-coated regions are visualized under RhB channel of the microscope, and FITC-HRP-coated regions are visualized under FITC channel of the microscope. Fluorescence images under RhB channel and FITC channel are collected, respectively, and merged into one. To obtain the three RhB-GOx-coated regions, region R1 is treated with $0.5 \mu\text{L}$, 10 mg/mL RhB-GOx solution; region R2 is treated with $0.5 \mu\text{L}$, 4 mg/mL RhB-GOx solution; region R3 is coated with $1 \mu\text{L}$, 1 mg/mL RhB-GOx solution. The left HRP-coated region is treated with $0.5 \mu\text{L}$, 1 mg/mL FITC-HRP aqueous solution, and the right region is coated with $1 \mu\text{L}$, 4 mg/mL FITC-HRP aqueous solution. The scale bar is 2 mm . (iii) RhB-GOx-coated regions treated with different concentration of RhB-GOx solutions have different fluorescence intensities.

The normalized fluorescence intensities are obtained by analyzing fluorescence microscope images under RhB channel.

Fig.5 d (i) Schematic showing an enzyme-mediated cascade reaction to decompose glucose, which can be visualized by a red fluorescence probe. (ii) Schematic and (iii) fluorescence microscope images showing the midstream decomposition of glucose into D-gluconolactone. The midstream channel with a GOx-coated region and an HRP-coated region converts glucose and Amplex Red into D-gluconolactone and resorufin, respectively. The scale bar is 1 mm.

Comment 12: Perfusion system: what is the composition of the gelation agent used to seal the needle to the channel?

Reply: 20 wt% gelatin aqueous solution is used to seal the joint between the channel and injection needle. The solution can solidify when allowed to stand under room temperature for a few minutes. The sealing process with gelatin solution is presented in Figure R3.

Figure R3. (i) A 20 wt% gelatin aqueous solution is applied to the channel-needle joint at 37°C, and allowed to solidify and seal the joint at room temperature. (ii) When a 2 mm wide channel is perfused with water at a flow rate of 0.5-3.0 ml/hour, no leakage from the sealing point is observed for at least 3 hours. The scale bar is 1 cm.

We are grateful for the reviewers' valuable and constructive comments, which have significantly improved the manuscript. We hope that the revisions make the manuscript acceptable for publication in *Nature Communications*.

REVIEWER COMMENTS

Reviewer #1 (Remarks to the Author):

The authors addressed most of the previous questions in a thorough manner. However, the exploration of the system in a wide range of diameters would improve the quality of the manuscript. By changing the inner diameter of the needles and flow rates, the relatively straightforward achievement of lower sized channels would be expected. The manuscript would much benefit from the establishment of the lower possible diameter achievable with the system. Additionally, the inner diameters (and not only external) of the utilized needles must be provided.

Reviewer #2 (Remarks to the Author):

The authors have undertaken extensive revisions, resulting in a marked improvement in the manuscript.

1. Part c in Figure 14 (SI) demonstrates a relationship between the allowable flow rates and the channel thickness. The provided range for the channel width, and consequently the flow rates, is wide. Could the authors derive a governing dimensionless number based on the channel properties (e.g., thickness, wall characteristics), fluid properties, and flow conditions? This could guide the relationship between the maximum allowable flow rates and channel thickness.
2. The authors should conduct a thorough proofreading of both documents (manuscript and SI) to ensure they are flawless. For instance, in the SI, do the dots in Eq. 2 have any specific meaning? What about the boldface fonts in Eqs. 3 and 4? The description of parameters in Eq. 4 should not be included in the equation line.

Point-by-point response letter to reviewer's comments for manuscript:

Manuscript Number: NCOMMS-23-14375A

Vascular network-inspired fluidic system (VasFluidics) with spatially functionalizable membranous walls

Reviewer #1

Comment: The authors addressed most of the previous questions in a thorough manner. However, the exploration of the system in a wide range of diameters would improve the quality of the manuscript. By changing the inner diameter of the needles and flow rates, the relatively straightforward achievement of lower sized channels would be expected. The manuscript would much benefit from the establishment of the lower possible diameter achievable with the system. Additionally, the inner diameters (and not only external) of the utilized needles must be provided.

Reply: We thank the reviewer for appreciating our work and providing helpful suggestions. As suggested,

1. we have added the inner and external diameter of the utilized printing nozzle tips. Details on the nozzle diameter are now included in the Supplementary Note 2: “*the size of the print nozzle tip (inner diameter = $389.9 \pm 4.5 \mu\text{m}$, external diameter = $851.3 \pm 27.3 \mu\text{m}$), and the average value of its inner and outer diameters is $620.6 \mu\text{m}$a fine stainless steel needle (inner diameter = $129.5 \pm 1.5 \mu\text{m}$, external diameter = $245.9 \pm 1.7 \mu\text{m}$).*” We have also added figures of nozzle tips in Supplementary Fig. 1 and 12.
2. we have also printed thinner channels (width $\approx 200 \mu\text{m}$) with smaller printing nozzle (inner diameter = $129.5 \pm 1.5 \mu\text{m}$, external diameter = $245.9 \pm 1.7 \mu\text{m}$), as presented in Supplementary Fig. 12.

Supplementary Figure 1. Setups for printing channels. A 0.5-10 μL plastic pipette tip is used as a printing nozzle, which is fixed to a 3D printer and connected to a syringe pump. The scale bar is 1 mm.

Supplementary Figure 12. Printing of thinner channels with smaller printing nozzle. We customized (a) a printing nozzle with a stainless steel needle as nozzle tip, and printed (b) a channel with width of around 200 μm . (c) Confocal laser scanning microscope images showing cross sections of 3 different sized channels printed with the stainless steel nozzle. All scale bars are 200 μm .

Reviewer #2

General Comment: The authors have undertaken extensive revisions, resulting in a marked improvement in the manuscript.

Reply: We are grateful to the reviewer for providing helpful suggestions to improve our manuscript. Please find below our point-by-point responses to the questions:

Comment 1: Part c in Figure 14 (SI) demonstrates a relationship between the allowable flow rates and the channel thickness. The provided range for the channel width, and consequently the flow rates, is wide. Could the authors derive a governing dimensionless number based on the channel properties (e.g., thickness, wall characteristics), fluid properties, and flow conditions? This could guide the relationship between the maximum allowable flow rates and channel thickness.

Reply: We thank the reviewer for this important comment. In response, we have added one more note regarding the derivation of the dimensionless number based on the channel size as the Supplementary Note 3; we have also calculated the dimensionless number under different flow rates (Supplementary Fig. 15(c)):

Supplementary Note 3: A dimensionless number to guide possible flow rates in different sized channels

To guide the relationship between the maximum allowable flow rates and the channel size, we derive a governing dimensionless number χ based on the channel size. We approximate the channel cross section as a circle with a diameter of d . We model flow in the channel as a pressure driven pipe flow, where the inlet flow rate results in a flow pressure on the channel wall. The maximum flow rate is reached when the flow-induced shear pressure on the channel wall exceeds the bonding strength between the channel and the substrate.

Under flow rate Q , the shear pressure on the channel wall is

$$\tau_{wall} = \frac{128\mu Q}{4\pi d^3}$$

where μ is fluid viscosity.

The flow induced axial pressure P leads to the expansion of the soft channel, with the diameter expanding from d_0 to $d_0 + \Delta d$. For elastic materials, Δd correlates with d_0 via the Young's modulus E

$$\Delta d = \frac{P}{E} d_0$$

At the maximum flow rate above which the channel will detach from the substrate, the shear stress on the channel wall reaches an equilibrium with the channel-substrate adhesion strength, which gives

$$\frac{128\mu Q_{max}}{4\pi(d_0 + \Delta d_{max})^3} = E_{bond}$$

Assume $E_{bond} = \beta E_0$ and $\Delta d_{max} = \alpha d_0$, where α and β are prefactors, and E_0 is the unit adhesion strength between chitosan and Petri dish.

Here we have

$$\frac{128\mu Q_{max}}{4\pi(d_0 + \alpha d_0)^3} = \beta E_0$$

Or

$$\frac{128\mu Q_{max}}{4\pi\beta(1 + \alpha)^3 E_0 d_0^3} = 1$$

Note that the prefactors α and β could be constant, or dependent on the channel diameter d . In any case, we introduce a shape factor function for the channel $\lambda(d) = \beta(1 + \alpha)^3$. And we derive a dimensionless number that measures the ratio between the flow-induced shear stress on the channel wall and the adhesion strength of the channel on the substrate

$$\chi = \frac{128\mu Q}{4\pi\lambda E_0 d_0^3}$$

At the maximum flow rate of $Q \sim Q_{max}$ ($Q/Q_{max} \sim 1$), we have

$$\chi = \frac{128\mu Q}{4\pi\lambda E_0 d_0^3} \sim 1$$

It is difficult to accurately estimate $\lambda(d)$ and E_0 , as the channel-substrate interaction is complex and their contacting area could be composed of multiplayer of polymers. Nevertheless, we next consider a model case to show the dimensionless parameter χ is effective.

Model case:

Considering a weak binding between the channel and the substrate (or between chitosan and Petri dish) with $E_0 = 10$ mPa, we choose a shape factor function of

$$\lambda(d) = A \left(\frac{d}{d^*} \right)^\gamma + B$$

where d^* is a characteristic length scale.

The best fitting of the experimental data results in that $A = 199.8$, $B = -143.2$, $d^* = 0.56$ (mm) and $\gamma = 0.5$, as plotted in the following figure.

In this way, we can calculate χ under different flow conditions, as listed in Supplementary Fig.15c. Apparently, we have $0(0.1) \leq \chi \leq 0(1)$.

To further verify that χ guides the possible flow rates in VasFluidic channels of different sizes, we estimated the maximum flow rate for a channel with a width of 1.2 mm. The calculated maximum flow rate (924 mL h⁻¹) with $\chi = 1$ is close to the experimental value (1100-1300 mL h⁻¹), below which the channel wall remain intact, with no liquid leakage.

(c)

Channel Type	Channel width (mm)	Channel cross-sectional area (mm ²)	Maximum flow rates (mL h ⁻¹)	χ
Straight channel; 50-60 mm in length	0.6-0.7	0.21	6-8	0.12
	1	0.51	700-1000	1.57-2.25
	1.5	1.01	1380-1680	0.62-0.76
	2	1.98	> 1680 *	> 0.25

* Maximum flow rate for our pump is around 1680 mL h⁻¹, and the flow rate higher than 1680 mL h⁻¹ is not applied here.

Supplementary Figure 15. Liquid perfusion of straight channels with different widths. 0.001-0.005 wt% rhodamine 6G aqueous solution is used as perfusion liquids for clear visualization. Channel outlets are exposed to air for smooth flow of internal liquids. The channel widths are

measured before removing the printing matrix, corresponding to w shown in Supplementary Fig.10 (a). (a) Liquid perfusion into a channel with width of $610\ \mu\text{m}$. The channel walls detach from the substrate as the flow rate reaches about $8\ \text{mL h}^{-1}$. The scale bar is $5\ \text{mm}$. (b) Liquid perfusion into a channel with width larger than $1\ \text{cm}$. The liquid flow rate is larger than $20\ \text{mL h}^{-1}$. The scale bars is $5\ \text{mm}$. (c) Maximum flow rates for $50\text{-}60\ \text{mm}$ long straight channels with different widths, above which the channel walls will detach from the substrate. Channels are perfused under certain flow rates for at least $1\ \text{min}$ to observe if liquid will leak out of the channel, or the channel walls will detach from the substrate. χ is a governing dimensionless number related to allowable flow rates in different sized channels, as explained in Supplementary Note 3.

Supplementary Figure 10 (a). Morphologies of cross-sections are approximated as parts of a circle with a radius of r . The cross-sectional area, the height, the maximum width, the length of the APAM/chitosan membrane part, and the length that channel attached to the substrate are defined as a , h , w , m and s , respectively.

Comment 2: The authors should conduct a thorough proofreading of both documents (manuscript and SI) to ensure they are flawless. For instance, in the SI, do the dots in Eq. 2 have any specific meaning? What about the boldface fonts in Eqs. 3 and 4? The description of parameters in Eq. 4 should not be included in the equation line.

Reply: Thank you for the careful reading. As suggested, we have deleted the dots in Eq.2, unbolded Eq.3 and 4, and excluded the description from the equation line. Besides, we have double-checked descriptions in main manuscript and supplementary materials.

REVIEWERS' COMMENTS

Reviewer #1 (Remarks to the Author):

The authors fully addressed the questions raised in the previous revision rounds. I recommend the publication of the manuscript.

Reviewer #2 (Remarks to the Author):

The authors addressed previous comments and the manuscript is acceptable.

Point-by-point response letter to reviewer's comments for manuscript:

Manuscript Number: NCOMMS-23-14375B

Vascular network-inspired fluidic system (VasFluidics) with spatially functionalizable membranous walls

Reviewer #1

Comment: The authors fully addressed the questions raised in the previous revision rounds. I recommend the publication of the manuscript.

Reply: We are very thankful to all suggestions raised, which have greatly improved the quality of our manuscript.

Reviewer #2

Comment : The authors addressed previous comments and the manuscript is acceptable.

Reply: We appreciate the reviewer for the many helpful and high-quality suggestions in improving our manuscript.